

# Chemistry Across Multiple Phases (CAMP) version 1.0: An integrated multi-phase chemistry model

Matthew L Dawson[1,4], Christian Guzman[1], Jeffrey H Curtis[2,3], Mario Acosta[1], Shupeng Zhu[5], Donald Dabdub[6], Andrew Conley[4], Matthew West[3], Nicole Riemer[2], and Oriol Jorba[1]

[1]Barcelona Supercomputing Center, Barcelona, Spain
[2]Department of Atmospheric Sciences, University of Illinois at Urbana-Champaign, Urbana, Illinois, USA
[3]Department of Mechanical Science and Engineering, University of Illinois at Urbana-Champaign, Urbana, Illinois, USA
[4]Atmospheric Chemistry Observations and Modeling Laboratory, National Center for Atmospheric Research, Boulder, Colorado, USA
[5]Advanced Power and Energy Program, University of California, Irvine, California, USA
[6]Department of Mechanical and Aerospace Engineering, University of California, Irvine, California, USA

**Correspondence:** Oriol Jorba (oriol.jorba@bsc.es) and Nicole Riemer (nriemer@illinois.edu)

**Abstract.** A flexible treatment for gas- and aerosol-phase chemical processes has been developed for models of diverse scale, from box models up to global models. At the core of this novel framework is an "abstracted aerosol representation" that allows a given chemical mechanism to be solved in atmospheric models with different aerosol representations (e.g., sectional, modal, or particle-resolved). This is accomplished by treating aerosols as a collection of condensed phases that are implemented according to the aerosol representation of the host model. The framework also allows multiple chemical processes (e.g., gas- and aerosol-phase chemical reactions, emissions, deposition, photolysis, and mass-transfer) to be solved simultaneously as a single system. The flexibility of the model is achieved by (1) using an object-oriented design that facilitates extensibility to new types of chemical processes and to new ways of representing aerosol systems; (2) runtime model configuration using JSON input files that permits making changes to any part of the chemical mechanism without recompiling the model; this widely used, human-readable format allows entire gas- and aerosol-phase chemical mechanisms to be described with as much complexity as necessary; and (3) automated comprehensive testing that ensures stability of the code as new functionality is introduced. Together, these design choices enable users to build a customized multiphase mechanism, without having to handle pre-processors, solvers or compilers. Removing these hurdles makes this type of modeling accessible to a much wider community, including modelers, experimentalists, and educators. This new treatment compiles as a stand-alone library and has been deployed in the particle-resolved PartMC model and in the MONARCH chemical weather prediction system for use at regional and global scales. Results from the initial deployment to box models of different complexity and MONARCH will be discussed, along with future extension to more complex gas–aerosol systems, and the integration of GPU-based solvers.

## 1 Introduction

Decades of progress in identifying increasingly complex, atmospherically relevant mixed-phase physicochemical processes have resulted in an advanced understanding of the evolution of atmospheric systems. However, this progress has introduced





a level of complexity that few atmospheric models were originally designed to handle. Most regional and global models comprise a collection of chemistry and 'chemistry-adjacent' software modules (e.g., those for gas-phase chemistry, gas–aerosol partitioning, surface chemistry, condensed-phase chemistry, partitioning between condensed phases, cloud droplet formation, cloud chemistry, emissions, deposition, and photolysis) along with 'support' modules that calculate parameters needed by these

modules (e.g., vapor pressures, Henry's law constants, and activity coefficients). Software modules have, in most cases, been developed independently and with a focus on computational efficiency that often leads to significant development efforts when modules are coupled for the first time, or changes to the underlying mechanisms are implemented. Efforts have been made to standardize Earth science module integration (Jöckel et al., 2005). However, these typically retain the stand-alone nature of individual modules.

Because of the complexity of atmospheric aerosol systems, the treatment of gas and condensed-phase chemical processes is often compartmentalized into a number of sub-modules. When rates for processes occurring in, e.g., the gas phase and aqueous cloud droplets are similar, this compartmentalization can affect the accuracy of simulations (Nguyen and Dabdub, 2003). In addition, when several equilibrium-based schemes are employed, their coupling is not always straightforward, particularly when the systems they describe are related, as is the case for separate inorganic and aqueous organic modules, both of which

can affect pH and water activity. A fully integrated framework is therefore needed for the treatment of mixed-phase chemical processes with scalable complexity and applicability to various representations of aerosol systems (e.g., modal, sectional, or particle-resolved). Such a framework remains to be developed and a first step toward such a comprehensive system is the focus of this paper.

In this work we present Chemistry Across Multiple Phases (CAMP) version 1.0. CAMP is designed to provide a flexible

framework for incorporating chemical mechanisms into atmospheric host models. CAMP solves one or more mechanisms composed of a set of reactions over a time-step specified by the host model. Reactions can take place in the gas phase, in one of several aerosol phases, or across an interface between phases (gas or aerosol). CAMP is designed to work with any aerosol representation used by the host model (e.g., sectional, modal, or single particle) by abstracting the chemistry from the aerosol representation. A set of parameterizations may also be included to calculate properties, such as activity coefficients, needed

to solve the chemical system. CAMP is intended to couple to a variety of external solvers, including those designed for GPU accelerators. CAMP v1.0 has been coupled to a CPU-based solver to demonstrate its ability to solve multi-phase chemistry for a variety of aerosol representations.

Flexible codes for use in atmospheric chemistry modeling have been the focus of several developments in the community. The implementation and extension of gas-phase mechanisms involve significant effort for complex systems such as Earth Sys-

tem models. Therefore, chemical pre-processors have been developed to ease the modification of gas-phase mechanisms that are included in various Earth System models. The Kinetic PreProcessor (KPP) (Damian et al., 2002) has been widely used as a tool to generate gas-phase chemical mechanisms, for example with the Master Chemical Mechanism (MCM) (Saunders et al., 2003; Jenkin et al., 2003) and the Regional Atmospheric Chemistry Mechanism gas-phase chemistry mechanism (RACM) (Stockwell et al., 1997), in both box models (Knote et al., 2015; Sander et al., 2019) and chemical transport models such as

the ECHAM/MESSy Atmospheric Chemistry (EMAC) model (Jöckel et al., 2010), the Global 3-D chemical transport model





for atmospheric composition (GEOS-Chem) (Bey et al., 2001), the Weather Research and Forecast model WRF-Chem (Grell et al., 2005), the LOTOS-EUROS model (Manders et al., 2017) and the MONARCH model (Badia and Jorba, 2015; Badia et al., 2017). Similar to KPP, the GenChem chemical pre-processor is used for the EMEP MSC-W chemical transport model (Simpson et al., 2012) and the CHEMMECH pre-processor is used for the Community Multiscale Air Quality Modeling Sys-

tem (CMAQ) (USEPA, 2020). These pre-processors convert lists of input gas-phase chemical species and gas-phase reactions to differential equations in Fortran code.

Going beyond the gas phase, the Aerosol Simulation Program (ASP) (Alvarado, 2008) is an aerosol model that uses ASCII files for specifying the parameters for the chemical mechanism, aerosol thermodynamics, and other inputs, which are read once at the beginning of the simulation. ASP is written in Fortran and uses a sectional aerosol representation with the number of

sections adjustable at runtime. The ASP model can be used as a box model, to simulate a plume in the ambient atmosphere or a smog-chamber experiment and can be called as a subroutine within spatially-resolved models (Alvarado et al., 2009; Lonsdale et al., 2020).

More recently, several atmospheric chemistry box models written in languages other than Fortran have become available. When written in interpreted languages, such models do not require the use of compilers, which makes them easier to use. For

example, KinSim (Peng and Jimenez, 2019) is an Igor-based chemical gas-phase kinetics simulator that is used for teaching and research purposes. The PyBox model (Topping et al., 2018) is written in Python. PyBox reads in a chemical equation file and then creates files that account for the gas-phase chemistry and gas-to-particle partitioning using the UManSysProp informatics suite (Topping et al., 2016). PyBox is the basis for PyCHAM, a Python box model for simulating aerosol chambers (O'Meara et al., 2021) and for JlBox (Huang and Topping, 2020), a high-performance community multi-phase atmospheric

0D box-model, written in Julia. JlBox simulates the chemical kinetics of a gas phase and a fully coupled gas–particle model with dynamic partitioning to a fully moving sectional size distribution. JlBox also uses chemical mechanism files to provide parameters required for multi-phase simulations.

The common goal of these models is that the code can be used and modified easily by specifying the chemical mechanism, which may include multi-phase reactions, in easy-to-modify text files. This is also one of the design principles of CAMP, which

uses JSON files to define the chemical mechanism. However, CAMP goes beyond this in that it is designed to treat the gas phase and organic/inorganic aerosol phases as a single system, it provides easy portability across different aerosol representations, and it allows the full, multi-phase system to be configured at runtime. CAMP is designed to be used in box models and within 3D models, as we will demonstrate in this paper.

For the user, this means that there is no pre-processor involved. Instead, the JSON files can be updated to change the

chemical mechanism at runtime, for example by adding more chemical species, more reactions, or different kinds of reactions. This does not require recompiling the code and enables rapid testing and sensitivity analyses. Furthermore, it is easy to change the underlying aerosol representation (bins, modes or particle-resolved), which is helpful to assess structural uncertainty due to aerosol representation assumptions and to adjust the computational burden depending on the application. Another important consideration is that at a time when computer architectures evolve rapidly, only a one-time back-end change is needed when

the code is ported to a new machine, rather than a complete rewrite of the model for each new architecture.





We envision the development of CAMP to proceed in three phases. Phase 1 is described in this paper and consists of a proof-of-concept flexible multi-phase chemistry package for multiple aerosol representations operating within a box model and a 3-D regional model, prior to any optimization efforts. Phase 2 will address optimization issues, including the use of GPUs and multi-state solving, targeting the use of CAMP in large-scale models on modern computer architectures. Phase 3 will refactor the code based on lessons learned during Phases 1 and 2, with a focus on easier porting to different solvers and architectures.

The overall description of the CAMP framework is presented in Sect. 2. A fundamental feature of CAMP is its applicability to a wide range of host models. We describe the coupling of CAMP with models of different complexity in Sect. 3. The runtime configuration is demonstrated in Sect. 4 with examples of solving the same multi-phase chemical system using different aerosol representations and different host models. Section 5 presents concluding remarks and a future vision for CAMP.

## 2 Software design

CAMP has been designed to separate the specification of multi-phase chemical mechanisms from the implementation of specific solvers, and to be usable by a variety of host models. A high-level picture of how CAMP interacts with a host model and a solver, and how this differs conceptually from more traditional implementations, is illustrated in Fig. 1. In the traditional approach (Fig. 1a), individual model components are typically introduced into an atmospheric model by adapting their code to interact with the host model's infrastructure and method for describing the model state. Solvers are typically inseparable from the representation of the chemical system, and configurations of individual model components are often hard-coded, making the addition of new species and chemical processes difficult, particularly when these involve an aerosol phase. In contrast, CAMP is compiled as a library and exposes an application programming interface (API) that is used by a host model to initialize CAMP for a particular multi-phase chemical system, update rates for processes such as emissions or photolysis that are typically calculated in separate modules, and solve the multi-phase chemical system at each time step (Fig. 1b). On the back-end, CAMP is designed to interact with a variety of external solver packages, thus separating the specification of the chemical system from the solver.

CAMP has been designed for extensibility to a variety of solver strategies, including GPU-based solvers as illustrated in Fig. 2. It has also been designed for scalability of chemical complexity through use of a standardized JSON format for specifying multi-phase chemical systems at runtime (Sect. 2.3), and applicability to a variety of aerosol representations (e.g., modal, sectional, particle-resolved; Sect. 2.2.3). Here we describe CAMP version 1.0, which uses the CPU-based CVODE solver from the SUNDIALS package (Cohen et al., 1996) with a configuration described in Sect. 2.4.1, as an initial implementation.

The scalability and extensibility of CAMP is achieved through use of an object-oriented design, and particularly the abstraction of various components of the chemical system described throughout this section. Although object-oriented design has been around for decades, its adoption in atmospheric models has been slow. We therefore provide here a quick description of some terminology for readers who may not be familiar with object-oriented design (for further background see, e.g., Mitchell (2005) or Jacobson (1992)). An 'object' is a set of data together with functions that operate on that data. For example, we will store the reaction process "$O_3 + NO \rightarrow NO_2 + O_2$" as an object. The data in this object is a list of reactants and a



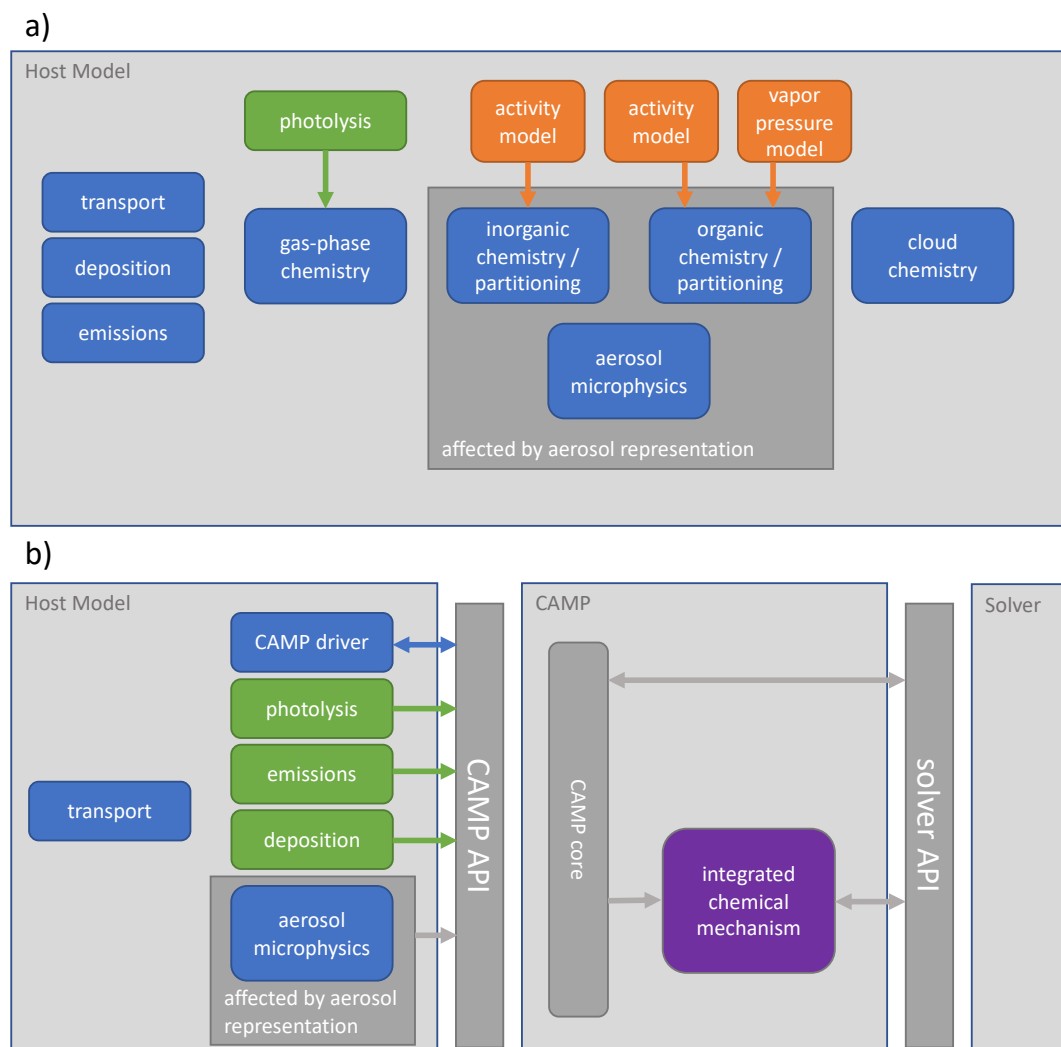

**Figure 1.** Interactions of chemistry and related modules in (a) a typical atmospheric model and (b) an atmospheric model using CAMP. Model components calculate rates or rate constants for physicochemical processes (green), calculate physical parameters (orange), or directly update the host model state (blue). Some modules that typically directly update the model state—deposition and emissions in (a)—now provide rates for these processes to CAMP (b). Parameter calculations—activity and vapor pressure models in (a)—are now integrated into the combined chemical mechanism (purple). Arrows indicate the primary flow of information among components.





list of products, and the object has a function that can compute the rate of change of each species given current conditions.

Objects are stored as variables in the code, so we could have an array of process objects, each of which describes a different chemical reaction. Every object belongs to a 'class', also called 'type', which specifies a minimal set of data and functions that the object must have. For example, we will have a `Process` class, which specifies that all objects of this type must have a `calculate_derivative_contribution()` function to compute the rate of change of each species. Classes can be organized into a hierarchy, with a 'base' class that is 'extended' by more specific classes. For example, we will use

`Process` as a base class and extend it by `Arrhenius` and `Troe` classes, which will add functions that are specific to those types of reactions. Our "$O_3 + NO$" object would be of type `Arrhenius` and thus implicitly also have type `Process`. The advantage of organizing code in this way is that other subroutines don't need to know about the details of different processes. For example, the time stepper code can simply take an array of `Process` objects and treat them all the same by calling their `calculate_derivative_contribution()` functions, without needing to know which of them are actually of type

`Arrhenius` or `Troe`.

A full description of the advantages of object-oriented programming is beyond the scope of this article, but the extensibility of a code to new problems through abstraction of key software components is of particular benefit to science models. In Sect. 2.2 we describe in detail how CAMP uses generalized base classes. We will use this language-independent terminology where possible throughout this paper to focus on the structure rather than the implementation of CAMP.

**2.1 CAMP Interface**

The general process for adding CAMP to a model is to create an instance of the `CampCore` class during model initialization, passing it a path to the configuration data. Each instance of the `CampCore` class is configured for one particular chemical mechanism. Sets of `CampCore` objects can be created for solving multiple chemical mechanisms. The `CampCore` object acts as the interface between the host model and CAMP, handling requests to update species concentrations, rates, rate con-

stants, and other mechanism parameters, solve the chemical system for a given time step, and retrieve updated chemical species concentrations. `CampCore` objects can also pack and unpack themselves onto a memory buffer for parallel computing applications.

**2.2 Abstraction of a chemical mechanism**

At the core of the chemistry package is an abstract chemical mechanism made up of instances of classes that extend one of three

base classes: `Process`, `Parameter`, and `AerosolRepresentation`. This approach is well-suited to physicochemical systems where components of the system must provide similar information about the current state of the system during solving, but where they apply different algorithms to calculate this information. In the following sections we describe the functionality of each of the three base classes, the classes that extend them, and their use.

Some class and function names have been changed here compared to their names in CAMP v1.0 for clarity of their purpose,

and will be updated in the next release of the code. The current naming scheme and detailed descriptions of CAMP v1.0 software components are described in the CAMP documentation.



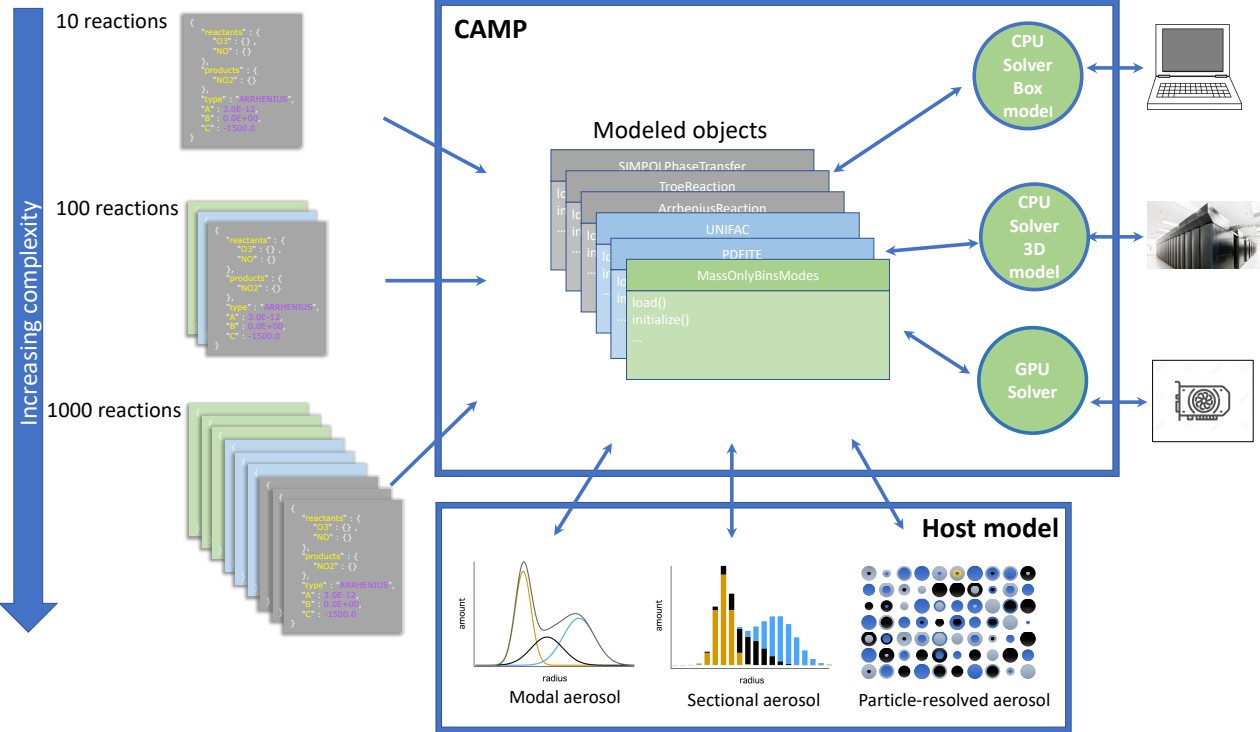

**Figure 2.** Schematic overview of the CAMP framework. The multi-phase chemical mechanism is flexibly defined using standardized JSON files. This is converted into an internal representation of model objects. This interfaces with the aerosol representation (modal, sectional, particle-resolved) determined by the host model, and can be coupled to solvers that are appropriate for the compute requirements of the application (CPU or GPU solvers on platforms ranging from personal computers to high-performance supercomputers.)

### 2.2.1 Processes

The primary responsibility of instances of classes that extend the `Process` base class is to provide contributions to the rates of change of chemical species (i.e., the forcing), and to the Jacobian of the forcing, if this is required by the solver, from a single physicochemical process. These processes include gas- and condensed-phase chemical reactions, the condensation and evaporation of condensing species, surface reactions, and any other processes that lead to changes in the state of a chemical species over time that are included in the CAMP mechanism. The functions of the `Process` class are shown in Table 1. A description of `Process`-extending classes is included in Table 2.

The Jacobian of the forcing calculated by the set of `Process` objects that make up a chemical mechanism includes parameters (described below) as independent variables. This allows the Jacobian for the solver (a square matrix including only solver variables) to be calculated as described in Sect. 2.2.2.





**Table 1.** Functions of the `Process` base class and its extending classes.

| `Process` function | Description |
|---|---|
| `get_used_jacobian_elements` | During initialization, the process indicates which Jacobian elements it will contribute to during solving. (This gives the solver the option of using a sparse Jacobian maxtrix.) |
| `update_ids` | During initialization, the solver generates IDs for species to use when providing contributions to their rates of change, and for Jacobian elements, for use during solving. |
| `update_for_new_environmental_state` | During solving, this function is called whenever the environmental state of the system (temperature, pressure, etc.) changes. This allows each process to re-calculate environmental state-dependent parameters (e.g., rate constants) only when necessary. |
| `calculate_derivative_contribution` | During solving, this function is called to calculate the rate of change for each of the species that participates in the process based on the current state of the system and update the `Forcing` object passed to this function to account for these changes using the indices saved during the call to `update_ids`. |
| `calculate_jacobian_contribution` | During solving, this function is called to calculate the contribution to the Jacobian matrix from this process based on the state of the system and update the `Jacobian` object passed to this function to account for these contributions using the ids saved during the call to `update_ids`. |

All processes listed in Table 2 are regularly tested under simple sets of conditions as described in Sect. 2.5. However, only those marked with an asterisk in Table 2 have been evaluated thoroughly in the context of a comprehensive mechanism, as discussed in Sect. 4. Remaining processes are listed in Table 2 for completeness, but should be considered as being under development.

### 2.2.2 Parameters

`Parameter`-extending classes provide values for properties that can be diagnosed from the system state (e.g., activity coefficients). These properties are used by `Process`-extending classes to calculate their contribution to the forcing of the chemical species, and to the Jacobian of the forcing. An advantage of the abstract mechanism design is that these `Parameter`-extending classes can be as complex as needed for a given application. For example, a box model could apply a detailed activity coefficient calculation scheme as part of a `Parameter`-extending class, and a global model may apply a different, simplified scheme in another `Parameter`-extending class. No changes to the `Process`-extending classes that use these activity coefficients would be required. The functions of the `Parameter` class are shown in Table 3. A description of the extending classes is included in Table 4.





**Table 2.** Current implementations of the `Process` base class. Classes marked with asterisk are used in the CAMP implementation example presented in Sect. 4.

| `Process`-extending class | Description | References |
|---|---|---|
| `Arrhenius`* | A gas-phase Arrhenius-like reaction. Arrhenius-like rate constants are calculated with optional temperature and pressure dependent terms:<br><br>$$k = A\exp\left(-\frac{E_a}{k_b T}\right)\left(\frac{T}{D}\right)^B (1.0 + EP),$$<br><br>where $E_a$ is the activation energy (J), $k_b$ is the Boltzmann constant (J K$^{-1}$), $A$ [(cm$^{-3}$)$^{-(n_r-1)}s^{-1}$], $D$ (K), $B$ (unitless) and $E$ (Pa$^{-1}$) are reaction parameters, $n_r$ is the number of reactants, $T$ is temperature (K), and $P$ is pressure (Pa). | Finlayson-Pitts and Pitts (2000)<br>Byun and Ching (2019) |
| `AqueousReversible` | A reversible aqueous reaction defined by a reverse rate constant $k_r$ [(kg m$^{-3}$)$^{-(n_p-1)}s^{-1}$] and an equilibrium constant $K_{eq}$:<br><br>$$K_{eq} = A\exp\left(C\left(\frac{1}{T} - \frac{1}{298}\right)\right)$$<br><br>where $A$ [(kg m$^{-3}$)$^{(n_p-n_r)}$] and $C$ (K) are reaction parameters, $n_p$ is the number of products, $n_r$ is the number of reactants, and $T$ (K) is temperature. | |
| `CondensedPhaseArrhenius` | As in `Arrhenius`, but for condensed-phase reactions. Units are M for aqueous reactions or mol m$^{-3}$ otherwise. | |
| `CustomH2o2`* | A reaction with a specialized rate constant for HO$_2$ self-reaction:<br><br>$$k = k_1 + k_2[\text{M}],$$<br><br>where $k_1$ and $k_2$ are `Arrhenius` rate constants with $D = 300$ and $E = 0$, and M is any third-body molecule. | Yarwood et al. (2005)<br>Burkholder et al. (2019) |
| `CustomOhHno3`* | A reaction with a specialized rate constant for the reaction of OH and HNO$_3$:<br><br>$$k = k_0 + \left(\frac{k_3[\text{M}]}{1 + k_3[\text{M}]/k_2}\right),$$<br><br>where $k_0$, $k_2$ and $k_3$ are `Arrhenius` rate constants with $D = 300$ and $E = 0$, and M is any third-body molecule. | Yarwood et al. (2005)<br>Burkholder et al. (2019) |
| `Emission`* | A process that accounts for sources of gas-phase chemical species. Emission rates can be specified in the CAMP configuration or passed to a `CampCore` object at runtime if the emission rates vary during a simulation. | |





| `Process`-extending class | Description | References |
|---|---|---|
| `FirstOrderLoss`[*] | A process that accounts for first-order loss of gas-phase chemical species. First-order loss rate constants can be specified in the CAMP configuration or passed to a `CampCore` object at runtime if the loss rate constants vary during a simulation. These can be used, for example, for dry or wet deposition, or wall loss in simulations of laboratory experiments. | |
| `HenrysLawPhaseTransfer` | Henry's Law condensation and evaporation, defined by equilibrium rate constants of the form: $$H(T) = H(298\mathrm{K}) \exp\left( C\left( \frac{1}{T} - \frac{1}{298} \right) \right)$$ where $H(298\mathrm{K})$ is the Henry's Law constant at 298 K (M Pa$^{-1}$), $C$ is a constant (K), and $T$ is temperature (K). Condensation rate constants $k_c$ are calculated according to Zaveri et al. (2008) as: $$k_c = 4\pi r_{\mathrm{eff}} D_{\mathrm{g}} f_{\mathrm{fs}}(\mathrm{Kn}, \alpha)$$ where $r_{\mathrm{eff}}$ is the effective radius of the particles (m), $D_{\mathrm{g}}$ is the diffusion coefficient of the gas-phase species (m$^2$ s$^{-1}$) and $f_{\mathrm{fs}}(\mathrm{Kn}, \alpha)$ is the Fuchs-Sutugin transition regime correction factor (unitless), Kn is the Knudsen Number (unitless) and $\alpha$ is the mass accommodation coefficient. Mass accommodation coefficients ($\alpha$) are calculated using the method of Ervens et al. (2003) and references therein. | Ervens et al. (2003) Zaveri et al. (2008) |
| `Photolysis`[*] | The photolysis of a gas-phase chemical species. Photolysis rate constants can be specified in the CAMP configuration or passed to a `CampCore` object at runtime if the photolysis rate constants vary during a simulation. | |
| `SimpolPhaseTransfer`[*] | Vapor-pressure based condensation and evaporation based on the SIMPOL.1 vapor-pressure parameterization of Pankow and Asher (2008). Condensation rate constants are calculated as in `HenrysLawPhaseTransfer`. The SIMPOL.1 vapor pressure is then used to calculate the evaporation rate. | Pankow and Asher (2008) Ervens et al. (2003) Zaveri et al. (2008) |



| Process-extending class | Description | References |
|---|---|---|
| Troe* | A Troe (fall-off) reaction with rate constants of the form: $$k = \frac{k_0[\mathrm{M}]}{1 + k_0[\mathrm{M}]/k_{\mathrm{inf}}} F_{\mathrm{C}}^{(1 + 1/N[\log_{10}(k_0[\mathrm{M}]/k_{\mathrm{inf}})]^2)^{-1}}$$ where $k_0$ is the low-pressure limiting rate constant, $k_{\mathrm{inf}}$ is the high-pressure limiting rate constant, M is any third-body molecule, and $F_{\mathrm{C}}$ and $N$ are parameters that determine the shape of the fall-off curve, and are typically 0.6 and 1.0, respectively (Finlayson-Pitts and Pitts, 2000; Byun and Ching, 2019). $k_0$ and $k_{\mathrm{inf}}$ are `Arrhenius` rate constants with $D = 300$ and $E = 0$. | Finlayson-Pitts and Pitts (2000) Byun and Ching (2019) |
| WennbergNoRo2 | Branched reactions with one branch forming alkoxy radicals plus $NO_2$ and the other forming organic nitrates. The rate constants for each branch are based on an Arrhenius rate constant and a temperature- and structure-dependent branching ratio calculated as: $$k_{\mathrm{nitrate}} = \left(Xe^{-Y/T}\right)\left(\frac{A(T,[\mathrm{M}],n)}{A(T,[\mathrm{M}],n) + Z}\right)$$ $$k_{\mathrm{alkoxy}} = \left(Xe^{-Y/T}\right)\left(\frac{Z}{Z + A(T,[\mathrm{M}],n)}\right)$$ $$A(T,[\mathrm{M}],n) = \frac{2 \times 10^{-22}e^n[\mathrm{M}]}{1 + \frac{2 \times 10^{-22}e^n[\mathrm{M}]}{0.43(T/298)^{-8}}} 0.41^{\left(1 + \left[\log\left(\frac{2 \times 10^{-22}e^n[\mathrm{M}]}{0.43(T/298)^{-8}}\right)\right]^2\right)^{-1}}$$ where $T$ is temperature (K), [M] is the number density of air ($\mathrm{cm}^{-3}$), $X$ and $Y$ are Arrhenius parameters for the overall reaction, $n$ is the number of heavy atoms in the $RO_2$ reacting species (excluding the peroxy moiety), and $Z$ is defined as a function of two parameters $(\alpha_0, n)$: $$Z(\alpha_0, n) = A(293\mathrm{K}, 2.45 \times 10^{19}\mathrm{cm}^{-3}, n)\frac{(1 - \alpha_0)}{\alpha_0}$$ where $\alpha_0$ is an empirically determined base-line branching ratio. | Wennberg et al. (2018) |



| `Process`-extending class | Description | References |
|---|---|---|
| `WennbergTunneling` | Reactions with rate constant equations calculated as: $$k = A \exp\left(-\frac{B}{T}\right) \exp\left(\frac{C}{T^3}\right)$$ where $A$ is the pre-exponential factor $((\mathrm{cm}^{-3})^{-(n-1)}\mathrm{s}^{-1})$, $B$ and $C$ are parameters that capture the temperature dependence, and $n$ is the number of reactants. | Wennberg et al. (2018) |

**Table 3.** Functions of the `Parameter` base class and its extending classes.

| `Parameter` function | Description |
|---|---|
| `get_used_jacobian_elements` | During initialization, the parameter indicates which elements of a partial-derivative matrix it will contribute to during solving. (This permits the use of a sparse partial-derivative matrix.) |
| `update_ids` | During initialization, the solver provides an ID for the parameter, and for partial derivatives, for use during solving. |
| `update_for_new_environmental_state` | During solving, this function is called whenever the environmental state of the system (temperature, pressure, etc.) changes. This allows each parameter to re-calculate environmental state-dependent sub-parameters only when necessary. |
| `calculate` | During solving, this function is called to calculate the parameter based on the current state of the system and update the parameter array passed to this function using the index saved during the call to `update_ids`. |
| `calculate_jacobian_contribution` | During solving, this function is called to calculate the contribution to the partial-derivative matrix from this process based on the state of the system and update the `Jacobian` object passed to this function to account for these contributions using the ids saved during the call to `update_ids`. |

In addition to calculating the value(s) of the parameter(s) during solving, a `Parameter`-extending class must also provide the partial derivatives of the parameter with respect to the solver variables. These are assembled into a matrix of partial derivatives in which the dependent variables are all the calculated parameters used by the processes that make up the chemical mechanism and the independent variables are the solver variables. This matrix is used with the Jacobian calculated by the set of processes (which includes the dependence of the forcing of solver variables on calculated parameters) to calculate the Jacobian
that is returned to the solver (a square matrix that includes only the solver variables).

    The `Parameter`-extending classes shown in Table 4 are included here as examples of how this type of class fits into the overall CAMP design. These parameterizations are regularly tested under simple sets of conditions as described in Sect. 2.5. However, they have not yet been thoroughly evaluated in the context of a comprehensive chemical mechanism, and should therefore be considered as being under development.





**Table 4.** Current implementations of the `Parameter` base class. These are included here for reference and will be described in more detail and evaluated in a separate paper.

| `Parameter`-extending class | Description | References |
|---|---|---|
| `PdfiteActivity` | Calculates aerosol-phase species activities using a Taylor series to describe partial derivatives of mean activity coefficients for ternary solutions, as described in Topping et al. (2009). | Topping et al. (2009) |
| `UnifacActivity` | Calculates activity coefficients for aerosol-phase species based on the total aerosol phase composition using functional group contributions. | Marcolli and Peter (2005) |
| `ZSRAerosolWater` | Calculates the equilibrium aerosol water content based on the Zdanovski–Stokes–Robinson mixing rule in the following generalized format: $$W = \sum_{i=0}^{n} \frac{1000 M_i}{\mathrm{MW}_i \, m_i(a_w)}$$ where $M_i$ is the concentration of binary electrolyte $i$ with molecular weight $\mathrm{MW}_i$ and molality $m_i$ at a given water activity $a_w$ contributing to the total aerosol water content $W$. | Jacobson et al. (1996) Metzger et al. (2002) |

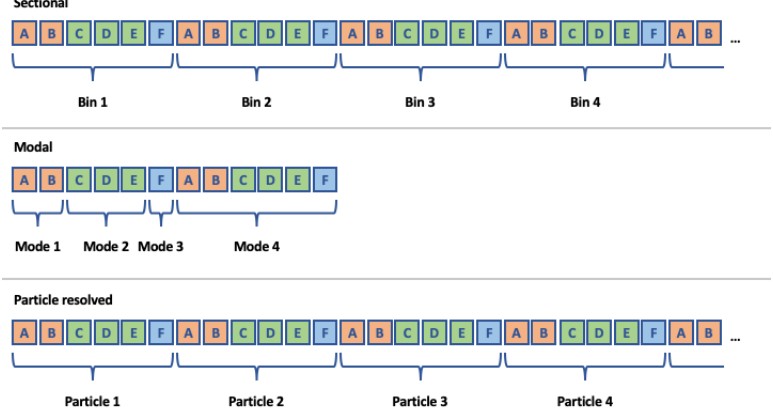

**Figure 3.** Common structures for aerosol state arrays in three different aerosol representations. Rows of boxes represent arrays used by a host model to describe the aerosol state during a simulation. Letters indicate unique condensed-phase chemical species concentrations. Colors indicate unique aerosol phases.





### 2.2.3  Aerosol Representations


As discussed in the introduction, software packages designed to account for aerosol processes (nucleation, coagulation, condensation/evaporation, condensed-phase chemistry, etc.) are often tightly coupled to the way the model represents aerosols (sectional, modal, or particle-resolved). Therefore, the addition of new condensed-phase species, reactions, and evaporation/condensation typically involves non-trivial modifications of the model code. A primary goal of CAMP is to treat multi-

phase chemical systems (including condensed-phase chemistry and evaporation/condensation) in models with different aerosol representations without the need for custom development when new species, reactions, and evaporation/condensation processes are added. Abstraction of these aerosol representations is how CAMP achieves this goal.

Abstraction of an aerosol representation requires a re-conceptualization of how aerosols are represented in models. Figure 3 shows common structures for three unique ways of representing aerosols in models: sectional, modal, and particle-resolved

approaches. Typically, each entity in the representation (bins, modes, or particles in the example shown in Fig. 3) includes a set of chemical species whose concentrations vary during the simulation. However, when gas-phase species condense onto an aerosol particle, they are typically assumed to condense into a specific 'phase' of the aerosol. For example, if a sectional model includes black carbon and a set of organic species in each section, the condensation of gas-phase organics would usually be calculated based on physical properties of the aerosol particles (e.g., size) and the chemical composition of the condensed-

phase organics—i.e., the black carbon is assumed to not be mixed with the organics. Although this concept of distinct phases of aerosol matter within individual particles is implicitly used in the development of contemporary aerosol models, they usually are not explicitly treated as such in the software. Thus, the first step in abstracting aerosol representations in CAMP is to introduce unique aerosol phases in the software using the `AerosolPhase` class.

Instances of the `AerosolPhase` class account for specific sets of chemical species that make up an aerosol phase (repre-

sented by colors in Fig. 3): an 'aqueous sulfate' `AerosolPhase` might include water, sulfate, nitrate, ammonia, etc., whereas a 'sea-salt' `AerosolPhase` might include these same species along with sodium and chloride. What makes these objects useful is that the condensed-phase chemistry and the set of species that condense into each `AerosolPhase` are the same for every instance of the phase that exists in a particular aerosol representation. For example, if the host model employs a sectional aerosol scheme with an 'aqueous sulfate' phase included in each bin, and an aqueous oxidation reaction is included in the

chemical mechanism for the 'aqueous sulfate' phase, this reaction is applied to the 'aqueous sulfate' phase in each section.

Although the specific condensed-phase species, reactions, and condensation/evaporation are the same for all instances of a particular `AerosolPhase`, the *rates* of reaction and of condensation/evaporation will differ among instances of a particular phase and often depend on physical properties of the aerosol particle in which the phase is present. Thus, the second step in abstracting aerosol representations is to define an `AerosolRepresentation` base class that defines functionality to provide

these properties based on the host model's scheme for describing aerosols. The functions of the `AerosolRepresentation` class are shown in Table 5. A description of the extending classes is included in Table 6.

The `AerosolRepresentation` base class allows specific aerosol representations to calculate the physical parameters of aerosol particles (number concentration, effective radius, etc.) using whatever algorithm applies to that scheme. The only



**Table 5.** Primary functions of the `AerosolRepresentation` base class and its extending classes. In addition to returning the property requested, each of these functions can also return the partial derivatives of the property with respect to the solver variables for use in calculating the Jacobian of the forcing.

| `AerosolRepresentation` function | Description |
|---|---|
| `effective_radius__m` | During solving, this function can be passed an instance of an aerosol phase and it will return the effective radius [m] of the particle(s) in which this phase exists. |
| `number_concentration__n_m3` | During solving, this function can be passed an instance of an aerosol phase and returns the number concentration [$\mathrm{m^{-3}}$] of the particle(s) in which this phase exists. |
| `aerosol_phase_mass__kg_m3` | During solving, this function can be passed an instance of an aerosol phase and returns the total mass concentration [$\mathrm{kg\,m^{-3}}$] of the phase. |
| `aerosol_phase_average_molecular_weight__kg_mol` | During solving, this function can be passed an instance of an aerosol phase and returns the average molecular weight [$\mathrm{kg\,mol^{-1}}$] of the phase. |

requirement is that it provides these properties during solving. For example, a single-moment mass-based sectional scheme may

have a fixed effective radius for each section and calculate the number concentration based on the total mass of each species in each phase that is present in the section, whereas a particle-resolved scheme may explicitly track number concentration but calculate the effective radius based on the total mass of each species in each phase that is present in the particle. A new class extending `AerosolRepresentation` must be introduced for each new scheme for representing aerosols—e.g., a two-moment mass- and number-based sectional scheme could be added as an `AerosolRepresentation`-extending

class—but once it is introduced into the CAMP code, any multi-phase chemical mechanism supported by CAMP can be used with the new aerosol scheme. In this paper, we have two `AerosolRepresentation`-extending classes, one that is called `MassBasedModalSectional` and another called `SingleParticle`. The class `MassBasedModalSectional` is set up to define modes or sections (or a combination of both) with fixed geometric mean diameters (GMD) and standard deviations (GSD) for modes, and fixed mid-point diameters for sections. This particular choice was made to replicate the

aerosol representation that is currently used in the MONARCH model, but additional `AerosolRepresentation` classes can be added to accommodate other types of modal or sectional representations, e.g., a two-moment mass- and number-based modal scheme with variable geometric mean diameters.

## 2.3 JSON mechanism description

Model element classes (described above) are designed to provide the structure of `Process`, `Parameter`, and

`AerosolRepresentation` calculations without being fixed for a particular set of model conditions. Model configura-





**Table 6.** Current implementations of the `AerosolRepresentation` base class. Note that many details of these aerosol representations (e.g., number of modes or sections, mode/bin GMD/GSD, number of computational particles) can be easily configured at runtime.

| `AerosolRepresentation`-extending class | Description | References |
|---|---|---|
| `MassBasedModalSectional` | A mass-based modal/sectional scheme with fixed geometric mean diameters (GMD) and standard deviations (GSD) for modes, and fixed mid-point diameters for sections. This can be used to support only modes or only sections, or to support a combination of modes and sections. | Spada (2015) |
| `SingleParticle` | A particle-resolved aerosol scheme in which the aerosol is represented as a representative sample (typically $10^3$–$10^6$ computational particles) of the total number of aerosol particles. The state of each particle is based on the mass of each species present in the particle, and the number of actual particles the computational particle represents. | Riemer et al. (2009) |

tion files must therefore be able to handle complex data structures (e.g., the functional group contributions or interaction maps required by `Parameter` calculations). We use the JSON format for model configuration files. JSON is a widely used format for semi-structured data, is human readable, and a large number of free tools are available for validating and interacting with JSON data. This structure allows chemical mechanisms to be fully runtime configurable. Importantly, the JSON structure

coupled with simple interactive tools allows users who are not experts in model development to easily simulate new chemical processes either in an isolated system (e.g., to simulate a flow-tube or chamber experiment) or as part of an existing comprehensive atmospheric chemical mechanism.

Figure 4 shows two examples of JSON configuration objects used by CAMP. The first is the relatively simple example of an Arrhenius reaction. The second is a portion of one of the more complex configuration data sets used in CAMP—that

of the UNIFAC activity model (Fredenslund et al., 1975). Note that the JSON format handles the complex structure of data representing functional group parameters and their interaction parameters without imposing artificial constraints on the number of functional groups or their interaction parameters. These complex data sets are typically hard-coded into model code, and require recoding whenever a new functional group or interaction is needed. The JSON format allows CAMP to access this data at runtime. As a result, users can easily modify the UNIFAC model by a simple change to the configuration files.





```
{
  "reactants" : {
     "O" : {} ,
     "NO2" : {}
  },
  "products" : {
     "NO" : {}
  },
  "type" : "ARRHENIUS",
  "A" : 5.6E-12,
  "B" : 0.0E+00,
  "C" : 180.0
}
```

(a) Arrhenius reaction

```
{
  "name" : "n-butanol",
  "type" : "CHEM_SPEC",
  "UNIFAC groups" : {
     "OH" : 1,
     "CH2(-OH)" : 1,
     "CH2(hydrophobic tail)" : 2,
     "CH3(hydrophobic tail)" : 1
  }
  ...
},
{
  "name" : "n-butanol/water activity",
  "type" : "SUB_MODEL_UNIFAC",
  "phases" : [ "n-butanol/water mixture" ],
  "functional groups" : {
     "CH2(-OH)" : {
        "main group" : "CHn(-OH)",
        "volume param" : 0.6744,
        "surface param" : 0.540
     },
     "CH2(hydrophobic tail)" : {
        "main group" : "CHn(hydrophobic tail)",
        "volume param" : 0.6744,
        "surface param" : 0.540
     },
     ...
  },
  "main groups" : {
     "CHn(-OH)" : {
        "interactions with" : {
           "OH" : 986.5,
           "H2O" : 2314
        }
     },
     "OH" : {
        "interactions with" : {
           "CHn(-OH)" : 156.4,
           "CHn(hydrophobic tail)" : 156.4,
           "H2O" : 276.4
        }
     },
     ...
  }
}
```

(b) UNIFAC activity model

**Figure 4.** Two examples of CAMP configuration data in JSON format: an Arrhenius reaction (a), and a portion of a UNIFAC activity model configuration (b). Ellipses (...) indicate portions of the data omitted for brevity.





## 2.4 Computational implementation

Solving the chemical system often accounts for a large fraction of the computational cost of atmospheric models (Christou et al., 2016). The primary goal of CAMP is to provide a means to configure a full mixed-phase chemical system at runtime, independent of the specific aerosol representation used by the host model. In its final form, it will provide an infrastructure for coupling external ODE solvers, which can be optimized for particular chemistry configurations and computational hardware.

In this section, we describe how the `Process`, `Parameter`, and `AerosolRepresentation` interfaces can be used to provide information needed by an external ODE solver.

### 2.4.1 External ODE solver

The design of CAMP allows the user to configure a variety of gas-phase, condensed-phase, or multi-phase chemical mechanisms. Regardless of the size or the degree of stiffness of the resulting system of differential equations, CAMP aims to obtain results for all cases while meeting user specifications of timestep error tolerance, order of solution approximation, and convergence tolerance, by eventually coupling to a suite of external solver packages.

In this first phase of development, we coupled CAMP to the external CVODE solver of the SUNDIALS package (Cohen et al., 1996) using the Backward Differentiation Formulas (BDFs) and Newton iteration. This algorithm is suitable for mathematically stiff systems. The variable-order, variable time-step CVODE solver with time-step error control provides accurate solutions, which is why it was chosen for this initial evaluation (Cohen et al., 1996). This algorithm requires the solution of a linear system at each time-step. We chose the KLU sparse solver of the SuiteSparse package, which for chemical systems typically requires less storage than the dense or banded solvers (Palamadai Natarajan, 2005).

### 2.4.2 Workflow and CAMP solving functions

Implicit integration of stiff ODEs requires the computation of both the forcing (rates of concentration changes) and the Jacobian of the forcing. As a result, CAMP computes the forcing as well as the analytical Jacobian of the forcing, placing the values in the data structures provided by the solver. This section describes the interactions among a host atmospheric model, CAMP, and an external ODE solver. Figure 5 illustrates the workflow during model initialization.

First, the user defines the chemical system in the JSON format described in Sect. 2.3. The host model initializes a `CampCore` object (Sect. 2.1) with the user-provided JSON files. During initialization, the `CampCore` creates the set of `Process`, `Parameter`, and `AerosolRepresentation` objects that describe the chemical system based on the JSON data.

After the `CampCore` is initialized, the host model has the option of forming connections to specific `Process` objects whose properties will be set from external modules (photolysis, emissions, deposition, etc.). These connections are returned to the host model as objects from the `CampCore`, which can be used at runtime by the host model to update `Process` parameters (e.g., photolysis or deposition loss rate constants, or emission rates). This allows host models to use modules external to CAMP for the calculation of rates and rate constants.



**Figure 5.** CAMP initialization workflow.



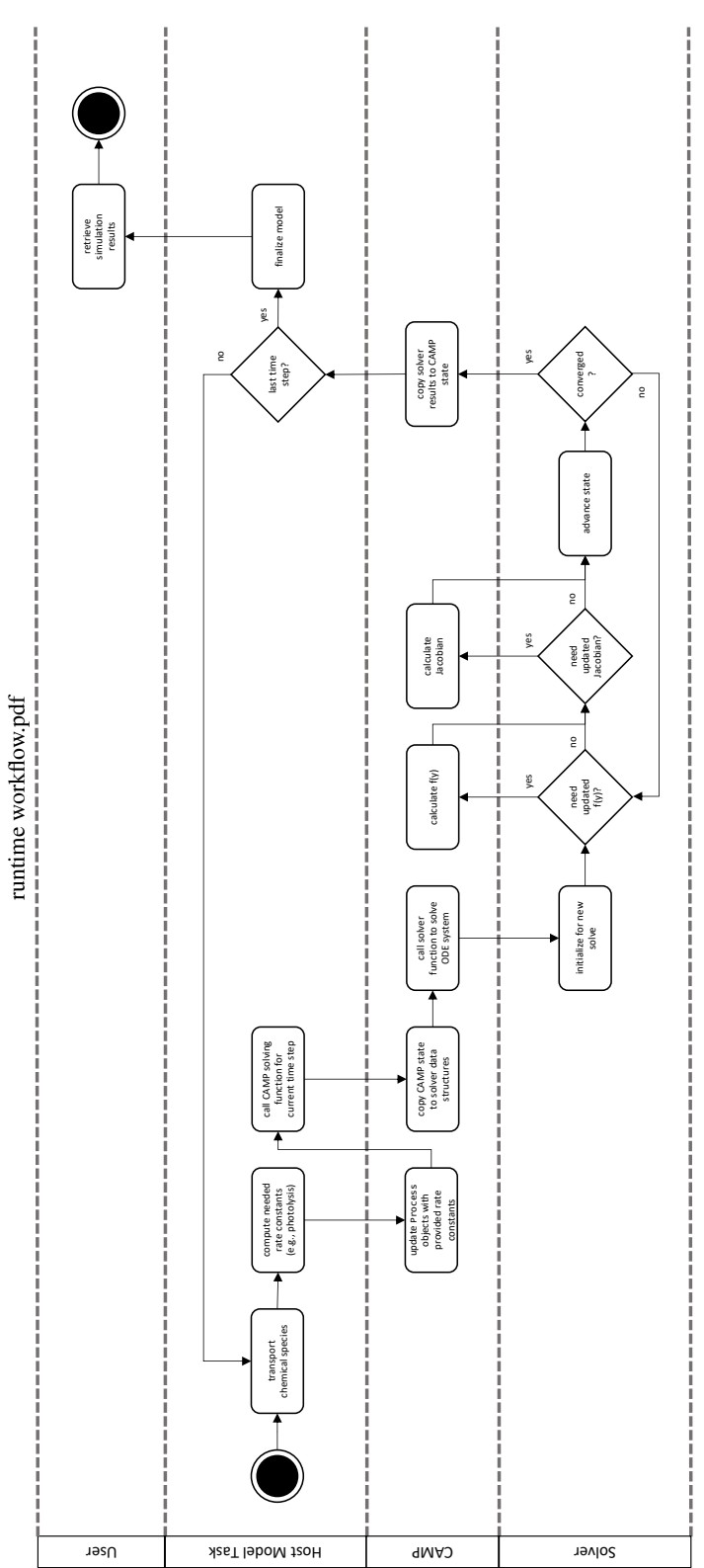

**Figure 6.** CAMP runtime workflow.



rate calculation workflow.pdf

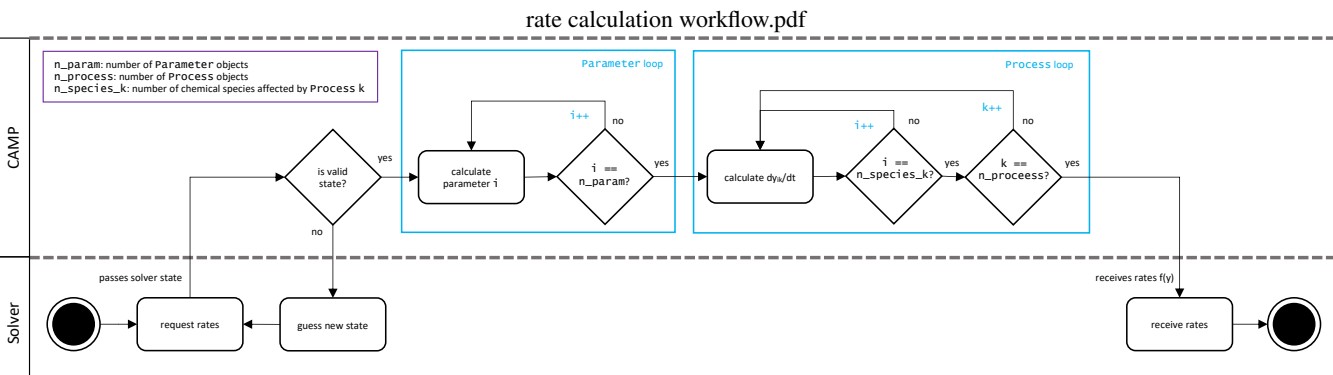

**Figure 7.** CAMP rate calculation workflow. Jacobian calculations follow a similar pattern.

In message passing interface (MPI) or threaded applications, the initialization described above can take place on the primary task. The initialized `CampCore` and any `Process`-connection objects would then be packed into a memory buffer, distributed to the secondary tasks, and unpacked into new objects for use during the model run. Once a `CampCore` object exists on each compute task, it is told to initialize the external ODE solver, including configuring it to use the CAMP functions that calculate

the forcing $f(y)$ and the Jacobian of the forcing.

During the simulation (Fig. 6), a host model iterates over its domain, using the `CampCore` to solve the chemical system for each discrete air mass. For each domain component, the host model uses its `Process`-connection objects to update any rates/rate constants for the current time step. It then passes the necessary state data (temperature, pressure, species concentrations, aerosol state data, etc.) along with the timestep over which to solve the chemical system to the `CampCore`.

When a `CampCore` is asked to solve chemistry for a given set of initial conditions and time step, it first transfers the state data into the data structures of the external ODE solver. It then instructs the external solver to solve the system of ODEs over the given time step. The ODE solver can call the CAMP functions that calculate $f(y)$ and the Jacobian of $f(y)$ for a particular set of conditions at any point during the solve.

The workflow of the CAMP function that calculates $f(y)$ is shown in Fig. 7, and the CAMP function that calculates the

Jacobian of $f(y)$ follows the same general workflow. Both functions iterate first over the collection of `Parameter` objects to calculate parameterizations for the current solver state, and then over the collection of `Process` objects, collecting contributions from each to either $f(y)$ or the Jacobian of $f(y)$ through functions of the `Process` interface (row 4 or 5 in Table 1). Thus, the forcing $f(y)$ for a particular species $i$ is calculated as

$$f_i \equiv \frac{dy_i}{dt} = \sum_k \left( \frac{dy_i}{dt} \right)_k ,$$





where $\left(\frac{dy_i}{dt}\right)_k$ is the forcing of species $i$ due to process $k$. Similarly, the partial derivative of the forcing of species $i$ with respect to species $j$ is

$$\frac{df_i}{dy_j} = \sum_k \left(\frac{df_i}{dy_j}\right)_k.$$

The way CAMP disentangles the specification of a multi-phase chemical system from the particular way aerosols are represented is by providing information needed by any particular `Process` related to aerosols through the `Aerosol-`

`Representation` interface. `Process` objects that affect or depend on aerosol species are always set up to actually operate on those species within a particular `AerosolPhase` (Sect. 2.2.3). The `Process` is applied equally to every instance of the `AerosolPhase`, whether that instance exists in a mode, or a section, or a single particle. A modal scheme may implement a phase once in a particular mode, or in several modes, and a sectional or particle-resolved scheme may implement this phase in every section or particle or in only certain sections or particles. In any of these situations, a `Process` that operates on a

particular `AerosolPhase` operates on each instance of the `AerosolPhase` as determined by the aerosol scheme.

When a `Process` needs information related to the particle(s) in which a particular phase exists, it accesses this information through the `AerosolRepresentation`. For example, as a Henry's Law processes is calculating contributions to $f(y)$ for a particular `AerosolPhase`, it calls the `effective_radius__m` function of the `AerosolRepresentation` to obtain the effective radius of the particle(s) in which the `AerosolPhase` exists. The way the aerosol scheme stores the dependent

data is hidden from, in this case, the Henry's Law `Process`, allowing aerosol schemes to be flexible in their underlying representation and the way they calculate, e.g., effective radii. The aerosol functions listed in Table 5 can also return the partial derivatives of the property they return with respect to the solver state variables. Thus, a `Process` is able to calculate its contribution to the Jacobian of $f(y)$, including the dependence of aerosol properties on state variables, without knowing specifically how the property is being calculated. In a similar way, parameterizations and their partial derivatives with respect

to state variables are accessed through the `Parameter` interface as described in Sect. 2.2.2.

After the ODE solver converges on a solution, the final state is returned to the `CampCore`, which in-turn returns it to the host model. The host model can then continue to the next time step.

## 2.5 Testing

When new code is pushed to the CAMP GitHub repository, an automated process (GitHub Actions) builds the library and runs

a suite of tests (both unit and integration tests) to ensure the new code does not break existing functionality. An attempt has been made to organize the CAMP source code into short, well-defined functions to which unit tests can be applied (generally tests of a single function with exact or nearly exact expected results). Integration tests (where the whole CAMP model is run under prescribed conditions) are also included, which consist of, e.g., simulations of the CB05 mechanism and comparison with results using a KPP-generated CB05 solver, and an Euler backward iterative solver (Hertel et al., 1993).

In addition to unit tests and integration tests of comprehensive chemical mechanisms, a series of integration tests for simple systems (comprising instances of only a single type of process or parameter) is run to test each `Process`- and `Parameter`-extending class. If possible, CAMP simulation results for the simple chemical systems used in tests are compared with an-





**Table 7.** Tests applied to `Process`- and `Parameter`-extending classes.

| Class | Test type | Test data reference |
|---|---|---|
| `Arrhenius` | Analytic solution[a] | |
| `AqueousReversible` | Analytic solution[a] | |
| `CondensedPhaseArrhenius` | Analytic solution[a] | |
| `CustomH2o2` | Analytic solution[a] | |
| `CustomOhHno3` | Analytic solution[a] | |
| `Emission` | Analytic solution[a] | |
| `FirstOrderLoss` | Analytic solution[a] | |
| `HenrysLawPhaseTransfer` | Approximate solution[b] | |
| `Photolysis` | Analytic solution[a] | |
| `SimpolPhaseTransfer` | Approximate solution[b] | |
| `Troe` | Analytic solution[a] | |
| `PdfiteActivity` | Comparison with hard-coded calculations for $H^+$–$NH_4^+$–$SO_4^{2-}$–$NO_3^-$ system described in equations 16 and 17 of Topping et al. (2009) | Topping et al. (2009) |
| `UnifacActivity` | Comparison with hard-coded calculations and published results for n-butanol/water system | Marcolli and Peter (2005) |
| `ZsrAerosolWater` | Comparison with hard-coded calculations for the NaCl and $CaCl_2$ systems using molality calculations from Jacobson et al. (1996) and Metzger et al. (2002). | Jacobson et al. (1996) Metzger et al. (2002) |

[a] Analytic solution tests run a simulation of a simple chemical system that can be solved analytically.

[b] Phase transfer tests run a simulation of a simple system with large initial particle mass relative to the mass available for transfer. Results are compared approximately assuming the mass transferred has only a small affect on the total particle mass, and thus on calculated uptake rates.

alytical solutions of the system. When systems that can be solved analytically could not be identified for tests of particular processes, approximate solutions are compared with the CAMP simulation results. For `Parameter`-extending classes, which
do not require solving, but whose calculations are complex and thus more error-prone, hard-coded calculations for specific published systems are compared to CAMP results for the parameterization. The types of tests performed for each `Process`- and `Parameter`-extending class are listed in Table 7.

## 3 Host models

A key feature of the CAMP framework is its applicability to atmospheric models with diverse ways of representing aerosol
populations. Thus, for this initial evaluation, two models that exist at opposite ends of the aerosol-representation spectrum are used as test beds for the CAMP framework. The MONARCH chemical weather prediction system employs a single-moment





mass-based representation of aerosols as a mixture of sections and modes (Spada, 2015). The particle-resolved PartMC model represents aerosol particles as a sample of discrete computational particles, each with a unique chemical composition and size (Riemer et al., 2009). To demonstrate the universal applicability of CAMP, after the CAMP framework was integrated

into these two models, the chemical gas-phase mechanism traditionally used by the MONARCH model was translated to the CAMP input file format (Sect. 2.3) and run in both PartMC and MONARCH. Two gas–aerosol partitioning reactions that form secondary organic aerosol (SOA) and that are part of the traditional MONARCH model were also added. Importantly, once CAMP was integrated into the MONARCH and PartMC models, the application of this specific mechanism required no changes to the source code of CAMP or either host model, and required no recompilation of the models. A brief description of

the MONARCH and PartMC models follows.

### 3.1 MONARCH atmospheric chemistry model

The Multiscale Online AtmospheRe CHemistry (MONARCH) model (Pérez et al., 2011; Haustein et al., 2012; Jorba et al., 2012; Badia and Jorba, 2015; Badia et al., 2017; Spada, 2015; Klose et al., 2021) is a fully online integrated system for meso- to global-scale applications developed at the Barcelona Supercomputing Center (BSC). The model provides operational re-

gional mineral dust forecasts for the World Meteorological Organization (WMO; https://dust.aemet.es/), and participates in the WMO Sand and Dust Storm Warning Advisory and Assessment System for Northern Africa-Middle East-Europe (http://sds-was.aemet.es/). Since 2012, the system has contributed global aerosol forecasts to the multi-model ensemble of the ICAP initiative (Xian et al., 2019) and since 2019, it has been a candidate model of CAMS—Air Quality Regional Production (https://www.regional.atmosphere.copernicus.eu).

A gas-phase module combined with a hybrid sectional–bulk multi-component mass-based aerosol module is implemented in the MONARCH model that uses the Nonhydrostatic Multiscale Model on the B-grid (NMMB; Janjic and Gall, 2012) as the meteorological core driver. The gas-phase scheme used in MONARCH is the Carbon Bond 2005 chemical mechanism (CB05; Yarwood et al., 2005) extended with chlorine chemistry (Sarwar et al., 2012). The CB05 mechanism is well formulated for urban to remote tropospheric conditions. It considers 51 chemical species, and solves 156 reactions. The photolysis scheme

used is the Fast-J scheme (Wild et al., 2000). It is coupled with physics of each model layer (e.g., aerosols, clouds, absorbers such as ozone), and it considers grid-scale clouds from the atmospheric driver. The aerosol module in MONARCH describes the lifecycle of dust, sea-salt, black carbon, organic matter (both primary and secondary), sulfate and nitrate aerosols (Spada, 2015). While a sectional approach is used for dust and sea-salt, a bulk description of the other aerosol species is adopted. A simplified gas–aqueous-aerosol mechanism accounts for sulfur chemistry. The production of secondary nitrate–ammonium

aerosol is solved using the thermodynamic equilibrium model EQSAM. A two-product scheme is used for the formation of SOA from biogenic gas-phase precursors. Meteorology-driven emissions are computed within MONARCH. Mineral dust emissions can be calculated using one of the schemes described in Pérez et al. (2011) and Klose et al. (2021), several source functions are available to compute sea salt aerosol emissions (Spada et al., 2013), and biogenic emissions use the MEGANv2.04 model (Guenther et al., 2006).





In this work, the model was configured for a regional domain covering Europe and part of northern Africa. A rotated
latitude–longitude projection was used, with a regular horizontal grid spacing of 0.2 degrees. The top of the atmosphere was
set at 50 hPa with 48 vertical layers. Figure 10a displays the domain of study. Meteorological initial and boundary conditions
were obtained from the ECMWF global model forecasts at 0.125 degrees (ECMWF, 2020) and chemical boundary conditions
from the CAMS global model forecasts at 0.4 degrees (Flemming et al., 2015). The applied anthropogenic emissions are based
on the CAMS-REG-APv3.1 database (Kuenen et al., 2014; Granier et al., 2019) and the biomass burning emissions (forest,
grassland and agricultural waste fires) are from the GFASv1.2 analysis (Kaiser et al., 2012). Both datasets were processed
using the HERMESv3 system, an open source, stand-alone multi-scale atmospheric emission modelling framework developed
at the BSC that computes gaseous and aerosol emissions for use in atmospheric chemistry models (Guevara et al., 2019).
The HERMESv3 system was used to remap the original datasets and to derive hourly and speciated emissions. Aggregated
annual emissions were broken down into hourly resolution using the emission temporal profiles reported by van der Gon et al.
(2011). The speciation of NMVOC and PM emissions was performed using the split factors reported by Kuenen et al. (2014).
The autosubmit workflow manager was used for efficient execution of the MONARCH modelling chain (Manubens-Gil et al.,
2016).

## 3.2  PartMC

PartMC is a stochastic, particle-resolved aerosol box model, which resolves the composition of many individual aerosol parti-
cles within a well-mixed volume of air. Riemer et al. (2009), DeVille et al. (2011), Curtis et al. (2016), and DeVille et al. (2019)
describe in detail the numerical methods used in PartMC. To summarize, the particle-resolved approach uses a large number
of discrete computational particles ($10^4$ to $10^6$) to represent the particle population of interest. Each particle is represented by
a "composition vector", which stores the mass of each constituent species within each particle and evolves over the course of
a simulation according to various chemical or physical processes. The processes of coagulation, particle emissions, dilution
with the background, and losses due to dry deposition are simulated with a stochastic Monte Carlo approach by generating a
realization of a Poisson process. The "weighted flow algorithm" (DeVille et al., 2011, 2019) improves the model efficiency and
reduces ensemble variance.

We initialized the simulations shown in this paper with $10^4$ computational particles. This number changes over the course
of the simulation due to particle emissions and particle loss processes, but is kept within the range of $5 \times 10^3$ and $2 \times 10^4$ by
"doubling/halving," which is a common Monte-Carlo particle modeling approach to maintain accuracy (Liffman, 1992). If the
number of computational particles drops below half of the initial number, the number of computational particles is doubled by
duplicating each particle; if the number of computational particles exceeds twice the initial number, then the particle population
is down-sampled by a factor of two. These operations correspond to a doubling or halving of the computational volume.

PartMC typically uses the Model for Simulating Aerosol Interactions and Chemistry (MOSAIC) (Zaveri et al., 2008) to
account for multi-phase chemical process. However, for this paper, the MOSAIC chemistry was disabled in PartMC, replaced
by the CAMP framework, and simulations were performed with coagulation disabled for easier comparison with the box model
runs that used sections and modes, as described in Sect. 4.1.





**Table 8.** Specification aerosol representation for the CAMP box model set up

| Name | Aerosol representation | Comments |
|---|---|---|
| CAMP-bins | 8 logarithmically spaced sections | Partitioning of secondary aerosol changes mass in sections, but mass is not transferred between sections. No coagulation |
| CAMP-modes | three log-normal modes | Partitioning of secondary aerosol changes mass in modes, but geometric mean diameters/standard deviation of modes does not change. No coagulation. |
| CAMP-part | 10 000 computational particles, poly-disperse distribution | Partitioning of secondary aerosol changes mass and size of particles. No coagulation. |

## 4  Results

### 4.1  CAMP box model set up

To evaluate the CAMP framework, we set up three box model simulations that shared the same gas-phase chemistry and aerosol–gas partitioning, but differed in their aerosol representation. The gas-phase chemistry was the CB05 mechanism with extended chemistry for chlorine (Yarwood et al., 2005; Sarwar et al., 2012), and photolysis reaction rates were kept constant in time. Gas-phase initial conditions and gas-phase emissions are listed in Table 10. Environmental conditions were set to an air temperature of 290 K and air pressure of 1000 hPa. The mechanism was further extended with secondary aerosol production from isoprene using the model as shown in Table 9. The partitioning of the isoprene products to the aerosol phase was allowed on primary and secondary organic aerosols.

The aerosol representations consisted of the following (Table 8): (1) "CAMP-modes" used three log-normal modes, (2) "CAMP-bins" used eight logarithmically spaced sections, and (3) "CAMP-part" used 10 000 discrete computational particles.

The initial aerosol distribution consisted of three log-normal modes (Table 11), and was taken from Seinfeld and Pandis (2016), Ch. 8. The CAMP-modes simulation was directly initialized with these three modes. For the CAMP-bin simulation, we discretized the three modes into 8 logarithmically spaced sections between 6.57 nm and 24.85 μm. The first section was defined at minus three standard deviations of the geometric mean diameter of the fine mode and the last section at plus three standard deviations of the geometric mean diameter of the coarse mode. The mass of the three modes was distributed accordingly into the eight sections. For the CAMP-part simulation, we sampled the initial aerosol distributions with 10 000 computational particles.

Only the CAMP-part simulations considered particle growth due to the condensation process of gas-phase precursors. The CAMP-modes and CAMP-bins representations mimic the approach taken in the MONARCH model (see Sect. 3.1), where effective radii and geometric standard deviations of the modes are fixed over the course of a simulation, and secondary aerosol mass does not move between sections, i.e., aerosol growth is not represented. In contrast, the CAMP-part representation does include aerosol growth. The microphysical process of aerosol growth could be easily included for the modal and sectional





representations if desired, and would not interfere with the already existing implementation of particle growth for the particle-resolved representation. Coagulation is not included in the CAMP-bins and CAMP-modes implementation. While coagulation is available in the CAMP-part simulations, it was disabled for the CAMP-part simulation to allow for an easier comparison of
440 all simulations.

**Table 9.** Gas–aerosol partition mechanism. Phase-transfer reactions are based on the SIMPOL.1 model calculations of vapor pressure described by Pankow and Asher (2008).

| gas-phase (mass-based stoichiometry) | | |
|---|---|---|
| reaction | rate constant | reference |
| ISOP + OH → 0.192 ISOP-P1 | $2.54 \times 10^{-11} \exp(407.6/T)$ | Yarwood et al. (2005); Tsigaridis and Kanakidou (2007) |
| ISOP + O$_3$ → 0.215 ISOP-P2 | $7.86 \times 10^{-15} \exp(-1912/T)$ | Yarwood et al. (2005); Tsigaridis and Kanakidou (2007) |

| gas-aerosol partitioning reactions and SIMPOL B parameters | | | |
|---|---|---|---|
| reaction | B1 | B2 | B3 and B4 |
| ISOP-P1 ⇌ SOA1(a) | $3.81 \times 10^3$ | $-2.13 \times 10^1$ | 0. |
| ISOP-P2 ⇌ SOA2(a) | $3.81 \times 10^3$ | $-2.09 \times 10^1$ | 0. |

$T$ stands for air temperature.

## 4.2 Box model results

The purpose of this section is to demonstrate that all three CAMP implementations yield the same results when given identical inputs. The results also reveal important structural differences between the modal implementation and the bin and particle-resolved representation. Starting with the example of a gas-phase species, Fig. 8(a) shows the simulated gas-phase mixing ratios
of O$_3$ for the three cases (CAMP-modes, CAMP-bins and CAMP-part) for the 24-hour simulation period. Since the CAMP modeling framework allows for flexibility in aerosol representation while maintaining an identical chemistry mechanism, the results for ozone mixing ratios are nearly identical for all three cases.

Figure 8(b) shows the evolution of gas-phase species involved in SOA formation: the precursor isoprene (ISOP) and the semi-volatile products in the gas phase, ISOP-P1 and ISOP-P2, where P1 is the product of ISOP reacting with OH and P2 is
450 the product of ISOP reacting with O$_3$. All three cases apply the same set of reactions, which yields the same production of SOA gas species. The particle-resolved and sectional case show somewhat higher ISOP-P1 mixing ratios compared to the modal case. Concurrently, the particle-resolved and sectional solutions for the ISOP-P1_aero mass concentration are comparable whereas the modal solution produces greater amounts of ISOP-P1_aero, shown in Fig. 8(c). The reason for the modal model





**Table 10.** Initial conditions and emission fluxes for gas-phase species for box model simulations.

| Gas species | Initial (ppb) | Emission rate (mol m$^{-3}$ s$^{-1}$) |
|---|---|---|
| NO | 0.1 | $1.44 \times 10^{-10}$ |
| NO2 | 1.0 | $7.56 \times 10^{-12}$ |
| HNO3 | 1.0 | |
| O3 | $5.0 \times 10^{1}$ | |
| H2O2 | 1.1 | |
| CO | $2.1 \times 10^{2}$ | $1.96 \times 10^{-9}$ |
| SO2 | 0.8 | $1.06 \times 10^{-9}$ |
| NH3 | 0.5 | $8.93 \times 10^{-9}$ |
| HCL | 0.7 | |
| CH4 | $2.2 \times 10^{3}$ | |
| ETHA | 1.0 | |
| FORM | 1.2 | $1.02 \times 10^{-11}$ |
| MEOH | $1.2 \times 10^{-01}$ | $5.92 \times 10^{-13}$ |
| MEPX | 0.5 | |
| ALD2 | 1.0 | $4.25 \times 10^{-12}$ |
| PAR | 2.0 | $4.27 \times 10^{-10}$ |
| ETH | 0.2 | $4.62 \times 10^{-11}$ |
| OLE | $2.3 \times 10^{-2}$ | $1.49 \times 10^{-11}$ |
| IOLE | $3.1 \times 10^{-4}$ | $1.49 \times 10^{-11}$ |
| TOL | 0.1 | $1.53 \times 10^{-11}$ |
| XYL | 0.1 | $1.40 \times 10^{-11}$ |
| NTR | 0.1 | |
| PAN | 0.8 | |
| AACD | 0.2 | |
| ROOH | $2.5 \times 10^{-2}$ | |
| ISOP | 5.0 | $6.03 \times 10^{-12}$ |
| O2 | $2.095 \times 10^{8}$ | |
| N2 | $7.8 \times 10^{8}$ | |
| H2 | $5.6 \times 10^{2}$ | |
| M | $1.0 \times 10^{9}$ | |



**Figure 8.** Comparison between CAMP-modes, CAMP-bins, and CAMP-part for (a) ozone mixing ratio, (b) ISOP, ISOP-P1 and ISOP-P2 mixing ratios, and (c) ISOP-P1_aero mass concentration for the 24-hour simulation period.



**Table 11.** Initial aerosol-phase conditions for box model simulations ("remote continental" case in Seinfeld and Pandis (2016), Ch. 8). POA stands for primary organic aerosol.

| Mode | Number concentration ($m^{-3}$) | Geometric mean diameter (m) | Geometric standard deviation | Composition |
|---|---|---|---|---|
| Aitken | $3.2 \times 10^8$ | $2.0 \times 10^{-8}$ | 1.45 | 100% POA |
| Accumulation | $2.9 \times 10^8$ | $1.16 \times 10^{-7}$ | 1.65 | 100% POA |
| Coarse | $3.0 \times 10^5$ | $1.8 \times 10^{-6}$ | 2.40 | 100% POA |

giving somewhat different results to the other two cases is that the rate of condensation is driven by particle size with smaller
particles reaching equilibrium more quickly than larger particles. The modal representation assumes one effective particle
radius for each of the three modes, while the sectional model assumes effective radii for each bin, and the particle-resolved
method tracks 10 000 individual particle diameters. The bin and particle methods more closely resemble one another and
therefore have similar results. They both represent larger particles resulting in ISOP-P1_aero condensing more slowly, and,
conversely, more ISOP-P1 remaining in the gas-phase for the CAMP-bins and CAMP-part cases (Fig. 8(b)). Since we can be
confident that all three simulations share the identical chemistry mechanism, we can attribute the differences entirely to the
aerosol representation.

## 4.3   3D Eulerian model results

As a final demonstration case, the 3D Eulerian model MONARCH was run using the CAMP framework to solve the same gas-
phase chemistry and gas-aerosol partitioning used in the box model simulations. The main difference between the MONARCH
configuration and the box models is the aerosol representation configuration. Using CAMP configuration files, only organic
aerosols were considered in the run with two primary modes, hydrophobic and hydrophilic, where the gas-aerosol partitioning
may occur. As described previously, the size of the mode is kept fixed during the simulation and no aerosol growth is considered.
A period of 20 days was simulated starting 21 July 2016 with initial concentrations of all gases and organic aerosols set to zero.
General model configuration details (i.e., domain, meteorology, chemistry, emissions and boundary conditions) are described
in Sect. 3.1.

Figure 9 shows the simulation results for $O_3$, ISOP and total isoprene SOA surface concentration at 12 UTC 9 August 2016.
Results are consistent with the box model runs, where regions with high $O_3$ and ISOP concentrations rapidly produce 0.5 to
5 $\mu g\ m^{-3}$ of SOA. This is particularly clear in central Portugal where biomass burning emissions inject large amounts of pri-
mary organic aerosols during the day. To provide some insights on the accuracy of the model results, surface $O_3$ concentrations
have been evaluated with observations of the European Environment Agency (EEA). The mean bias for all rural and urban-
background stations below 1000 m above sea level is shown in Fig. 10a for the period 28 July to 9 August 2016. Most stations
in Western and Central Europe have a bias below 5 ppbv. Figure 10b presents the time series of the EEA $O_3$ station-average.
Overall, model results are in reasonably good agreement with observations.



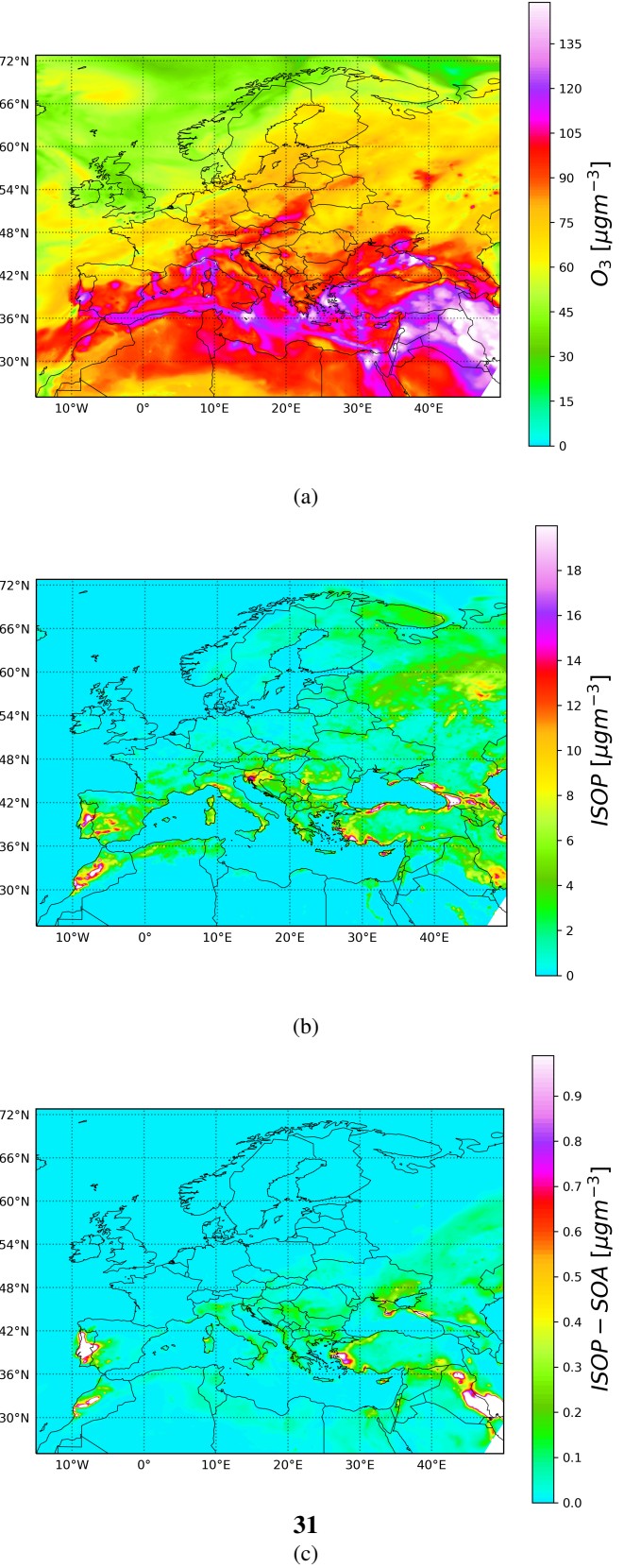

(c)

**Figure 9.** Surface concentration of (a) ozone, (b) isoprene and (c) total isoprene secondary organic aerosol for 9 August 2016 at 12UTC.







(a)

(b)

**Figure 10.** Evaluation of ozone surface concentration [ppbv] at the European Environment Agency (EEA) measurement sites: (a) Mean bias at rural and urban-background EEA sites below 1000 m above sea level for the period 28 July to 9 August 2016, (b) time series of ozone concentrations averaged over EEA sites (black dots: observations, red dots: model results).





# 5 Conclusions and future perspectives

## 5.1 Summary


This paper presents results from the first phase of a three-part development plan for CAMP: a flexible treatment for multi-phase chemistry in atmospheric models. The software package compiles as a library that can be linked to by models of various scale, from box models to regional/global atmosphere models. Gas- and condensed-phase chemistry along with evaporation/condensation, photolysis, emissions, and loss processes are solved as a single system, which is fully runtime-configurable

(i.e., no preprocessing or recompilation of code is necessary when modifying the chemical system). Importantly, this multi-phase chemistry treatment is independent of a host model's aerosol representation, such as modal, sectional, or particle-resolved schemes.

We demonstrate the applicability of CAMP for models that use different aerosol representations by coupling the CAMP library to the particle-resolved PartMC model as well as to the regional/global MONARCH model with a mixed modal/sectional

scheme. Box model results using modal, sectional, and particle-resolved aerosol schemes indicate that CAMP consistently solves the multi-phase chemical system for each aerosol representation. Differences in results for the time evolution of SOA formation between the modal representation on the one hand, and the particle-resolved and sectional representations on the other hand, can be entirely attributed to the chosen aerosol representation. Results from a regional MONARCH simulation over Europe are consistent with expectations and demonstrate that CAMP is applicable to large-scale atmospheric models.

Several design choices facilitate achievement of the product goals for CAMP:

- CAMP compiles into a stand-alone library; no modifications to the CAMP source code are necessary when porting to a new host model. This means that only a single CAMP code needs to be maintained, improving product sustainability.

- An object-oriented design, and specifically abstraction of physicochemical processes, diagnostic parameter calculations, and aerosol representations, allows CAMP to be extensible to new chemistry and physics, to be portable to models with

diverse ways of representing aerosol systems, and in its final form to be portable to a variety of solvers and computational architectures.

- Runtime JSON-based configuration eliminates the need for complicated preprocessing steps and recompilation of the model code when modifying the chemical system.

- A comprehensive testing strategy applying both unit and integration testing, automated using GitHub Actions continuous

integration, ensures the stability of the code as new chemical processes and aerosol representations are added.

Stability, portability to new models, and extensibility to new chemistry and physics are generally accepted as best practices for designing chemistry models. However, the runtime configurability of CAMP, which allows users to modify the chemical system without recompiling the model, has potential usefulness for a variety of applications where such changes are made frequently, such as data assimilation and sensitivity analyses. Additionally, runtime configuration means that CAMP can be

integrated into tools designed for users interested in simulating new or modified chemical systems who do not have a modeling





or software-development background. One such tool, an atmospheric chemistry box model with a browser-based interface for configuring, running, and analyzing results from the model, which uses CAMP to solve the chemical system, is currently being tested (https://github.com/NCAR/music-box).

## 5.2 Optimization, porting to GPUs, and future development

Development of CAMP is planned to occur in three phases. Phase 1 (this paper) entails a proof-of-concept library for solving multi-phase chemistry that is fully runtime configurable, applicable to models of various scale and ways of representing aerosols, and extensible to new physicochemical processes. In parallel with the planned development of the CAMP infrastructure, extension to new physicochemical processes will occur to support, e.g., aerosol surface reactions, deliquescence/efflorescence, and novel gas- and condensed-phase chemical reactions.

In Phase 2 (currently underway), the CAMP library is being coupled to a GPU-based ODE solver and optimized for large-scale models where efficiency is critical. For even moderately complex chemical mechanisms, solving the chemical system can account for a significant fraction of the computational expense of an atmospheric model. Thus, for CAMP to be suitable for weather, air quality, and climate models, efficient solving strategies are critical. Additionally, computational architectures evolve rapidly. Atmospheric models that are responsive to new hardware advances will provide more efficient, affordable

simulations and open the door to including more complex chemistry and physics that would otherwise be unfeasible. Thus, a key design goal of CAMP is to be portable to new solvers and computational architecture.

Preliminary results for Phase-2 work is available in Guzman-Ruiz et al. (2020). The optimized GPU-based strategy simultaneously solves multiple instances of a chemical system, represented in 3D models as grid cells or points. As part of the preliminary results, we compared a GPU version of the $f(y)$ function with an MPI simulation using the maximum number of physical

cores available in a node. The GPU version showed a computational time three times lower than the CPU-based MPI execution. The tests were performed on the CTE-POWER cluster provided by BSC (https://www.bsc.es/user-support/power.php). In addition, the final version of the GPU-based ODE solver is being designed for heterogeneous computing with CPUs. A detailed description of the methods is available in Guzman-Ruiz et al. (2020) and is expected to be presented in future publications.

Phase-3 development is planned as future work and will involve a refactoring of the code based on lessons learned in Phases 1

and 2, with a focus on improving the porting of CAMP to a variety of solving strategies and computational architectures.

*Code availability.* CAMP is available at https://github.com/open-atmos/camp. CAMP v1.0 is archived at https://doi.org/10.5281/zenodo. 5602154. The CAMP User Guide and BootCAMP Tutorial are available at https://open-atmos.github.io/camp. PartMC is available at https://github.com/compdyn/partmc. PartMC v2.6.0 is archived at https://dx.doi.org/10.5281/zenodo.5644422. The MONARCH code is available at https://earth.bsc.es/gitlab/es/monarch (last access: September 2021) (https://doi.org/10.5281/zenodo.5215467).



*Data availability.*  The source code and configuration JSON files for the modal and binned box model experiments are available in the CAMP repository (https://github.com/open-atmos/camp) in the data/CAMP_v1_paper folder. Box models and MONARCH outputs are available in https://doi.org/10.13012/B2IDB-8012140_V1.

*Author contributions.*  All authors contributed to writing, reviewing, and editing the draft. MD contributed to the conceptualization, design, and the development of the CAMP library. CG and MA contributed to the development of the CAMP library. JHC conducted box model

simulations. SZ contributed to testing the CAMP library. NR contributed to conceptualizing the paper, acquiring funding, and project administration. OJ contributed to conceptualizing the paper, acquiring funding, and conducted box model and MONARCH runs.

*Competing interests.*  The authors do not have any competing interests.

*Acknowledgements.*  MD received funding from the European Union's Horizon 2020 research and innovation programme under the Marie Skłodowska-Curie grant agreement no. 747048. MD, OJ, CG acknowledge the support from the Ministerio de Ciencia, Innovación y Univer-

sidades (MICINN) as part of the BROWNING project RTI2018-099894-B-I00. CG acknowledges funding from the AXA Research Fund. BSC co-authors also acknowledge the computer resources at MareNostrum and the technical support provided by Barcelona Supercomputing Center (AECT-2020-1-0007, AECT-2021-1-0027). NR, MW, and JHC acknowledge funding from grant NSF-AGS 19-41110. The National Center for Atmospheric Research is sponsored by the National Science Foundation. Any opinions, findings and conclusions or recommendations expressed in the publication are those of the author(s) and do not necessarily reflect the views of the National Science

Foundation



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
