# Peer review of "Chemistry Across Multiple Phases (CAMP) version 1.0: An integrated multi-phase chemistry model"

_Geoscientific Model Development, 2021_

## Author Comment (AC1)

**Response to the comments of Anonymous Referee #1 (RC1)**

We thank the reviewer for the constructive comments. Changes in response to the comments are marked in blue in the revised manuscript.

**(1.0)** Line 126: Instead of "Every object belongs to a class", consider "Every variable has a type, and the type of an object is its class. A class specifies..." Consider also breaking up this paragraph to emphasize the explanation of objects/classes and to highlight the examples.

> *We changed the text as suggested by the reviewer. It now reads (line 133): "As with all variables, every object has a 'type,' also called its 'class.' A class specifies...". We also broke up the paragraph to separate the examples of objects from those of classes.*

**(1.1)** Line 127: Regarding class hierarchies: the term hierarchy is used to describe a great many things. If you want to discuss polymorphism, consider this explanation or something like it: "Unlike other variables, objects have behaviors implemented by their functions. For example, chemicals reactions can be classified into Arrenhius and Troe types that share a common interface but have different implementations. This classification can be accomplished using a "base class" that defines the interface, and "subclasses" that implement different ways of satisfying it. So a Reaction base class could have Arrhenius and Troe subclasses that calculate reaction rates in their respective ways." Thereafter, you can refer to a subclass w.r.t. a base class for clarity. E.g. "Parameter subclasses" instead of "Parameter-extending classes".

> *We thank the reviewer for the good suggestion to use "subclass" rather than "...-extending class". We have changed this throughout the text. The paragraph in the introduction now reads (lines 133–138): "A class specifies a minimal set of data and functions that the object must have. For example, we will have a* `Process` *class, which specifies that all objects of this type must have a* `calculate_derivative_contribution()` *function to compute the rate of change of each species. Classes can be organized into a hierarchy, with a 'base' class that defines which functions must be supported, and 'subclasses' for specific implementations. For example, we will use* `Process` *as a base class with subclasses* `Arrhenius` *and* `Troe`*, which will implement those particular types of reactions."*

**(1.2)** Line 155: worth citing a link to CAMP's documentation here?

> *We added the citation (line 163).*

**(1.3)** Table 2: the equations for aqueous reversible reactions and for Henry's Law should each have a "K" after the 298 in the argument to the exp function.

> *We corrected this.*

**(1.4)** Line 243: Is it worth providing a reference or link that describes the JSON format and/or its use in software (ideally in a scientific context)?

> *We added a reference to the book: Introduction to JavaScript Object Notation: A To-the-Point Guide to JSON (line 248).*

**(1.5)** Figure 6: This is a very instructive figure. However, its orientation is a bit awkward for reading. To me, it seems like flipping it by 180 degrees might be better, if it doesn't go against journal guidance.

*We will check if this is possible with the typesetter.*

**(1.6)** Line 315: Consider, for clarity: "as determined by the aerosol scheme's representation."

*We updated the text as the reviewer suggests (lines 327–328)*

**(1.7)** Line 329: Consider providing a link to a description of GitHub Actions, since it's mentioned in a couple of places.

*We added a link to the GitHub Actions documentation (line 342).*

**(1.8)** Figure 8: the text in the legends for the plots does not display using two PDF renderers on my system.

*Figure 8 properly rendered in the submitted PDF version of the manuscript but did not properly render in the finalized version that appeared online. We have included the figure below as Figure 1.*

[Figure]

Figure 1: Comparison between CAMP-modes, CAMP-bins, and CAMP-part for (a) ozone mixing ratio, (b) ISOP, ISOP-P1 and ISOP-P2 mixing ratios, and (c) ISOP-P1_aero mass concentration for the 24-hour simulation period.

---

## Author Comment (AC2)

**Response to the comments of Anonymous Referee #2 (RC2)**

We thank the reviewer for the constructive comments. Changes in response to the comments are marked in blue in the revised manuscript.

**(2.0)** I wonder which aspects of an aerosol model are "outside of CAMP". This could be made more clear. I am wondering about advective & convective transport, vertical mixing, sedimentation, transport in rain drops, radiation interaction, fog formation, emissions, aerosol uptake of in cloud/ice water. Can fast and slow chemical reactions treated differently, to the advantage of computational efficiency?

> *We added a clarification beginning on line 113: "The current implementation of CAMP covers processes related to multiphase chemistry, i.e., gas-phase chemical reactions, heterogeneous reactions, gas-aerosol/cloud drop partitioning, aqueous-phase chemistry. Processes that are not part of multi-phase chemistry are "outside of CAMP." These include all transport processes (advection, turbulent diffusion), processes related to aerosol and cloud microphysics (e.g., formation of clouds, sedimentation of aerosol particles and cloud/rain drops/ice crystals, scavenging of aerosols by clouds, coagulation), emissions, and radiative processes. As indicated in Figure 1b, these processes are the responsibility of the host model and all related information needed by CAMP is communicated via the CAMP API."*

> *The solver takes care of treating chemical reactions, and we clarified this beginning on line 282: "Using the CVODE solver does not optimize for speed, as fast and slow chemical reactions are not treated differently. Future work, as part of the Phase-2 model development will focus on developing efficient solver strategies (see Section 5.2)"*

**(2.1)** A second question I have is the computational efficiency of CAMP. Is there any chance this can be quantified? And if not now, how can this be done in the future?

> *Although the computational efficiency is an interesting addition, it is outside the scope of this paper. Please notice that we will focus on the computational efficiency in a future work. This part would correspond to the Phase 2 of CAMP development, which is mentioned in the paper on line 94. In this future work, we plan to compare the performance with the rest of chemistry solver options integrated in MONARCH (KPP and a faster solver based on the Euler-Backward-Iterative solver method). Moreover, we are focusing on optimizing CAMP to reach competitive performance levels for pre-exascale machines, including the complete GPU porting. We are in the process of publishing a paper with these details.*

**(2.2)** Then - the chapter 4 is a bit disappointing. Shouldn't the box model and transport model simulations be compared to a classical code? Especially for MONARCH, there should be simulations available that are comparable.

> *The aim of Section 4 is to provide a proof-of-concept showing how CAMP can be used in models of different complexity. Although an in-depth evaluation and model intercomparison of the results shown here would be of interest, it is beyond the scope of the present manuscript.*

> *As described in Section 2.5, the CAMP codebase includes an extensive test suite that comprises both unit and integration testing. In addition to tests of individual CAMP Processes and Parameters (Table 7), CAMP is used to simulate the full CB05 mechanism under prescribed conditions. The results of this simulation are compared with those obtained using a KPP-generated Rosenbrock solver and an Euler-Backward-Iterative (EBI) solver for the same mechanism. Results for each species at each time step are compared to both the KPP and EBI results with a relative tolerance of $10^{-4}$ and an absolute tolerance of 1.0 ppb until concentrations drop below 1% of their initial*

[Figure]

Figure 1: Time series of ozone surface concentration [ppbv] at the European Environment Agency (EEA) rural and urban-background measurement sites below 1000 m above sea level for the period 28 July to 9 August 2016 (black dots: observations, red dots: MONARCH-CAMP model results, green dots: default MONARCH model results).

*values. These tests are run automatically every time new code is pushed to the CAMP repository. We believe that this automated testing provides confidence in the performance of the CAMP library. We have included more detail related to the tests in Table 7 and Appendix A. Please also see our response to comment 2.7.*

*As we mention in comment **(2.1)**, we plan to present in a future work a critical evaluation of the MONARCH-CAMP system together with a comprehensive discussion of differences and improvements obtained with a new chemistry solver like CAMP. Our aim in the **Results** section is to keep it concise and provide some show-case examples. In this sense, we prefer not to present an in-depth intercomparison with current default MONARCH model to avoid introducing a long discussion about the different model set up and configurations and how that impacts the results. However, following the suggestion of the reviewer we have introduced in the revised manuscript an evaluation of the default MONARCH model for the same period presented in the paper in comparison with MONARCH-CAMP results. We have updated Figure 10b of the manuscript with Figure 1 presented here. We show in the new figure a comparison of the surface concentration of ozone averaged over the European Environment Agency (EEA) rural and urban-background sites below 1000 m above sea level (black dots) compared with the MONARCH-CAMP run presented in the manuscript (green dots) and a MONARCH default version run (red dots). The MONARCH-CAMP results are consistent with the original MONARCH version and in good agreement with observations.*

*We have included the following text in the revised manuscript beginning on line 487: "Figure 10b presents the time series of the EEA $O_3$ station-average measurements (black dots) compared with results of a MONARCH-CAMP run (green dots) and the original MONARCH version described in Section 3.1 (red dots). Overall, the model results are in reasonably good agreement with observations. Differences between the two model runs may be attributed to the different treatment of some of the gas-phase chemistry related to the isoprene SOA production (reaction rates), the solver used (default MONARCH run uses an EBI solver), or missing heterogeneous chemistry processes in the MONARCH-CAMP set up (sulfate/nitrate/ammonium formation, hydrolysis of $N_2O_5$)."*

**(2.3)** Figure 8 in my pdf was not readable, legends disappeared, axis titles are absent, I dont understand what is shown. The discussion in 4.2 is too short to understand the result. Please rewrite.

*Figure 8 text did not render properly in the finalized PDF. We have included it in the response as Fig. 2.*

*Additionally we improved the reasoning behind the modal result diverging from the sectional and particle-resolved simulations. We have included the following text in the revised manuscript beginning on line 468, a new figure (shown here as Figure 3), and details in Appendix B:*

*"The reason for the modal model giving somewhat different results to the other two cases is that the rate of condensation is driven by particle size, with smaller particles reaching equilibrium more quickly than larger particles. This is illustrated in Fig. 9. This figure shows the characteristic timescale $\tau$ that is required to reach equilibrium (Zaveri et al., 2008). Appendix B lists the relevant equations to calculate $\tau$.*

*As indicated in Figure 9, the modal representation assumes one effective particle radius for each of the three modes (vertical broken blue lines), while the sectional model assumes effective radii for each bin (vertical dotted green lines), and the particle-resolved method tracks 10 000 individual particles radii (not shown). The bin and particle methods both represent larger particles compared to the modal method with correspondingly larger characteristic timescales to reach equilibrium resulting in ISOP-P1_aero condensing more slowly, and, conversely, more ISOP-P1 remaining in the gas-phase for the CAMP-bins and CAMP-part cases (Fig. 8(b)). Since we can be confident that all three simulations share the identical chemistry mechanism, we can attribute the differences entirely to the aerosol representation."*

*Appendix B: The characteristic timescale $\tau$ that is required to reach equilibrium (Zaveri et al., 2008) is defined as*

$$\tau = \frac{C_{a,i}}{|k_c(C_g - C^*)|}. \tag{1}$$

*Here, $C_{a,i}$ is the concentration of the condensed-phase product that resides in particle i with effective radius $R_i$, $C_g$ is the gas phase concentration of the same species, $C^*$ is the equilibrium gas-phase concentration with the particle phase (a function of temperature). The coefficient $k_c$ is the first-order mass transfer coefficient for the condensing gas given as*

$$k_c = 4\pi R_i D_g f(\text{Kn}, \alpha), \tag{2}$$

*where $D_g$ is the gas diffusivity, and $f$ is the transition regime correction factor as a function of Knudsen number and mass accommodation coefficient $\alpha$ defined as*

$$f(\text{Kn}, \alpha) = \frac{0.75\alpha(1 + \text{Kn})}{\text{Kn}(1 + \text{Kn}) + 0.283\alpha\text{Kn} + 0.75\alpha}. \tag{3}$$

**(2.4)** Figure 9 results: "regions with high O3 and ISOP concentrations rapidly produce 0.5 to 5 $\mu\text{g m}^{-3}$ of SOA. This is particularly clear in central Portugal where biomass burning emissions inject large amounts of primary organic aerosols during the day." I thought there are no primary aerosols in this CAMP version. How can BB primary emissions lead to SOA? Reconsider, rewrite please.

*In the first paragraph of Section 4.3, we present the aerosol representation used in the MONARCH-CAMP run, "Only organic aerosols were considered in the run with two primary modes, hydrophobic and hydrophilic, where the gas-aerosol partitioning may occur." Emissions of primary organic aerosol from both anthropogenic and biomass burning are considered as described in last paragraph of Section 3.1 and commented in line 470 of the manuscript. The SOA partitioning occurs both in anthropogenic and biomass burning POA particles. We added a clarification to line 487: "Note that both anthropogenic and biomass burning primary organic aerosol emissions are considered here."*

**(2.5)** Fig 10: Bias: "The mean bias for all rural and urban background stations below 1000 m above sea level is shown in Fig. 10a for the period 28 July to 9 August 2016." What is the value of this figure if the model is only computing SOA? I really dont understand what I can take from this figure and bias calculation.

[Figure]

Figure 2: Comparison between CAMP-modes, CAMP-bins, and CAMP-part for (a) ozone mixing ratio, (b) ISOP, ISOP-P1 and ISOP-P2 mixing ratios, and (c) ISOP-P1_aero mass concentration for the 24-hour simulation period.

[Figure]

Figure 3: Comparison between timescale $\tau$ for effective radii used by the modal and sectional aerosol representation. For reference, the initial aerosol mass distribution is shown.

*The MONARCH-CAMP model solves the same gas-phase chemistry (CB05) and the isoprene SOA partitioning used in the box model runs described in Section 4.1. This is mentioned in the first sentence of Section 4.3. In this sense, Figure 10 of the manuscript provides a synthesized evaluation of ozone as a key species in both the gas-phase chemistry mechanism and the SOA partitioning scheme solved by the model. As we have noted previously, we do not intend to provide a comprehensive evaluation of the model run in this manuscript, this will be presented in future works.*

*We added the following sentence to lines 407–408 to clarify that the full CB05 gas-phase mechanism is included as well as the SOA partitioning scheme: "The chemical processes included in the MONARCH-CAMP model set up consist of the full CB05 gas-phase mechanism coupled to a simple SOA formation mechanism, see Sect. 4.1 for details."*

**(2.6)** Ch 4.3 Why dont you show results of an ordinary MONARCH simulation for the same period compared to a CAMP-MONARCH simulation? Would be quite convincing if this would give similar results. At least for part of the chemical system.

*We followed the reviewer's suggestion. Please see our response in comment **(2.2)**.*

**(2.7)** Finally, I wonder if the tests in tab 7 could be explained, illustrated a bit more. How to quantify if the solution is comparable? Could this be discussed?

*We have added the criteria used to determine whether a test passes to Table 7. We have also included an appendix that describes the simple chemical systems used in many of the tests and the equations used to determine the analytical solutions to the evolution of these systems.*

**(2.8)** Conclusions: "Differences in results for the time evolution of SOA formation between the modal representation on the one hand, and the particle-resolved and sectional representations on the other hand, can be entirely attributed to the chosen aerosol representation." I am not sure this attribution is really demonstrated.

*We have added additional discussion in Section 4.2 to more clearly demonstrate that differences in the SOA partitioning is due to the difference in the aerosol respresentation. Please see our response in comment **2.3** for changes to manuscript.*

**(2.9)** "Results from a regional MONARCH simulation over Europe are consistent with expectations and demonstrate that CAMP is applicable to large-scale atmospheric models." What did you expect? what is demonstrated? please expand with a critical evaluation.

*Please see our response in comment **(2.2)** regarding the scope and extent of the evaluation results presented in this manuscript. The updated Figure 10b shows that MONARCH-CAMP results are consistent with previous versions of the model and evaluations. We have not detected any degradation of the results, that is what we were intending to highlight with "expectations and demonstration".*

*We have reformulated the sentence on lines 514–516: "Results from a regional MONARCH simulation over Europe are consistent with the previous model version and observations, and provide a proof-of-concept that CAMP is applicable to large-scale atmospheric models."*

**(2.10)** Table 3: I dont understand why the Parameter class has the same calculate jacobian contribution function as the Process class. The description text is just the same. Error, or - maybe I just dont understand.

*A `Process` that uses a `Parameter` to determine its contribution to the forcing of chemical species will have the `Parameter` object's Jacobian terms applied to its own Jacobian terms following the chain rule.*

*We have included an expanded description of this function in Table 3: "The partial derivatives returned from this function are applied to the Jacobian terms of any `Process` that uses this `Parameter` to determine its contribution to the forcing of chemical species following the chain rule."*

**(2.11)** Figure 4b: The entries into the json file are a cryptic for someone not knowing the UNIFAC model. How are users supposed to understand what needs to be changed within CAMP and what in json files?

*We have included text making clear the purpose of including this figure in the second paragraph of Section 2.3 beginning on line 256: "A full description of the JSON configuration format for the UNIFAC model is included in the CAMP documentation. We present a portion of this file in Figure 4 merely to illustrate two extremes in complexity of CAMP component configurations."*

*The CAMP library is designed such that users will not need to modify CAMP source code to introduce and configure new `Process` or `Parameter` objects. We have modified the sentence beginning on line 261 as follows:*

*"As a result, users can easily modify nearly every detail of the UNIFAC model by a simple change to the configuration files, without any need to modify the UNIFAC model source code. The specific parameters of the UNIFAC model that can be set in its JSON configuration file are described in the CAMP documentation."*

**(2.12)** Figure 6: what is a "CAMP state"?

*We have included a note in the figure caption to clarify this term: "Note that the CAMP state refers to the state of the coupled gas/aerosol system comprising temperature, pressure, gas-phase chemical species mixing ratios and the representation-specific state of the aerosol system. For*

*the single-particle representation, the aerosol state comprises the condensed species mass concentrations and the number of simulated particles the computational particle represents. For the modal/sectional representation, the aerosol state comprises the condensed species mass concentrations only.*"